psychology/behaviour

consensus decision-making, dyadic colour estimation, dynamic communication

**Author for correspondence:**
Da-Hui Wang
e-mail: wangdh@bnu.edu.cn

†These authors contributed equally to this study.

# Communication speeds up but impairs the consensus decision in a dyadic colour estimation task

Liutao Yu[1,2,†], Chundi Wang[3,1,†], Si Wu[4] and Da-Hui Wang[1]

[1]School of Systems Science and State Key Laboratory of Cognitive Science and Learning of China, Beijing Normal University, Beijing 100875, People's Republic of China
[2]School of Electronics Engineering and Computer Science, Peking University, Beijing 100871, People's Republic of China
[3]Department of Psychology and Research Centre of Aeronautic Psychology and Behavior, Beihang University, Beijing 100191, People's Republic of China
[4]School of Electronics Engineering and Computer Science, IDG/McGovern Institute for Brain Research, Peking-Tsinghua Center for Life Sciences, Peking University, Beijing 100871, People's Republic of China

D-HW, 0000-0002-6447-7516

Communication plays an important role in consensus decision-making which pervades our daily life. However, the exact role of communication in consensus formation is not clear. Here, to study the effects of communication on consensus formation, we designed a dyadic colour estimation task, where a pair of isolated participants repeatedly estimated the colours of discs until they reached a consensus or completed eight estimations, either with or without communication. We show that participants' estimates gradually approach each other, reaching towards a consensus, and these are enhanced with communication. We also show that dyadic consensus estimation is on average better than individual estimation. Surprisingly, consensus estimation without communication generally outperforms that with communication, indicating that communication impairs the improvement of consensus estimation. However, without communication, it takes longer to reach a consensus. Moreover, participants who partially cooperate with each other tend to result in better overall consensus. Taken together, we have identified the effect of communication on the dynamics of consensus formation, and the results may have implications on group decision-making in general.

# 1. Introduction

Group decision-making pervades our daily life, and communication between group members is known to play a key role [1–7]. Previous studies have shown that communication is a double-edged sword, which can lead to either good or bad consensus [1,3–6]. Granovskiy *et al.* showed that social influence can improve the answers to trivia questions [1]. Similar conclusions have recently been drawn by Jayles *et al.* [4] and Navajas *et al.* [5]. However, Lorenz *et al.* have shown negative effects of social influence using human experiments [6]. Madirolas and de Polavieja showed that resistance to social influence sometimes improves collective estimations [3]. Interestingly, Bahrami *et al.* showed that free oral communication and confidence sharing between similarly capable members lead to Bayesian optimal joint decisions in a perceptual discrimination task [8–12]. Miller and Steyvers showed that iterative communication between subjects can lead to better estimates in order-reconstructing tasks, which could be explained by a Bayesian model [13]. Voinov *et al.* demonstrated that perceptual localization judgements benefit from indirect interactions due to complementary information sharing [14]. However, Koriat directly chose the decision of the most confident member as the group decision and showed that a group decision is better than individual decisions when the majority are correct, without any communication [15,16]. However, despite the extensive findings from these studies, it is still unclear what is the exact role of communication in the consensus formation and how communication affects the consensus accuracy.

In this work, we address this issue by making use of a perceptual colour estimation task, modified from working memory tasks [17,18], in which a pair of isolated participants (one dyad) estimate the colours of discs presented on computer screens in front of them until they reach a consensus or complete eight estimations. We manipulated the presence of communication and the difference of stimuli to investigate the dynamic formation and accuracy of consensus in a dyad. We hypothesized that communication would speed up the consensus, and dyadic consensus would be better than individuals' reports. We also compared the quality of consensus in the with-communication and without-communication conditions. We found that communication between members speeds up consensus formation but impairs the accuracy.

# 2. Methods

## 2.1. Participants

This research was approved by the Institutional Review Board and Ethics Committee of Human Participant Protection under IRB no. 201905160043, School of Psychology at Beijing Normal University, Beijing, China. All participants provided informed consents and were compensated for their efforts. Each participant received RMB 60 yuan per hour. To encourage the participants to reach a consensus as accurately as possible, we provided RMB 100 yuan as a bonus to the top-ten dyads in each condition with regard to the accuracy of their consensus from trials in which they received the same stimulus. Whether reaching a consensus or not in one trial had no influence on their payments. This task had no requirement on response speed. In this research, we recruited 48 dyads from universities in Beijing to carry out the experiment. All participants were aged from 18 to 28. All participants had normal or corrected-to-normal vision, and normal colour vision as tested through the *PseudoIsochromatic Plate* (*PIP*) *Color Vision Test* [19]. Half dyads participated in the with-communication tasks, and the rest participated in the without-communication tasks (see section *Experimental procedure* for details). The sample size was determined from a power analysis (G*Power), which indicated that this number of participants was sufficient to detect a between-subjects effect (Cohen's $d = 0.964$) 90% of the time using a significance level of 0.05. The effect size was based on a similar collective perceptual decision-making study [8].

## 2.2. Experimental set-up

The experimental process was implemented through Psychtoolbox-3 based on Matlab 2015b [20–22]. The stimuli were presented on a 14-inch PC, the monitor of which had a resolution of 1366*768 pixels and a refresh rate of 60 Hz. Colours were described using HSV (hue–saturation–value) coordinates in this study, with the saturation and value kept at 100%. Therefore, different hues ($0° : 1° : 360°$) represented different colours, except that 0° and 360° represented the same colour here (see figure 1*a* for some examples). We arranged these hues in order and then obtained the colour wheel used to report (figure 1*b*). The colour wheel was randomly rotated across trials to avoid the memorizing effect.

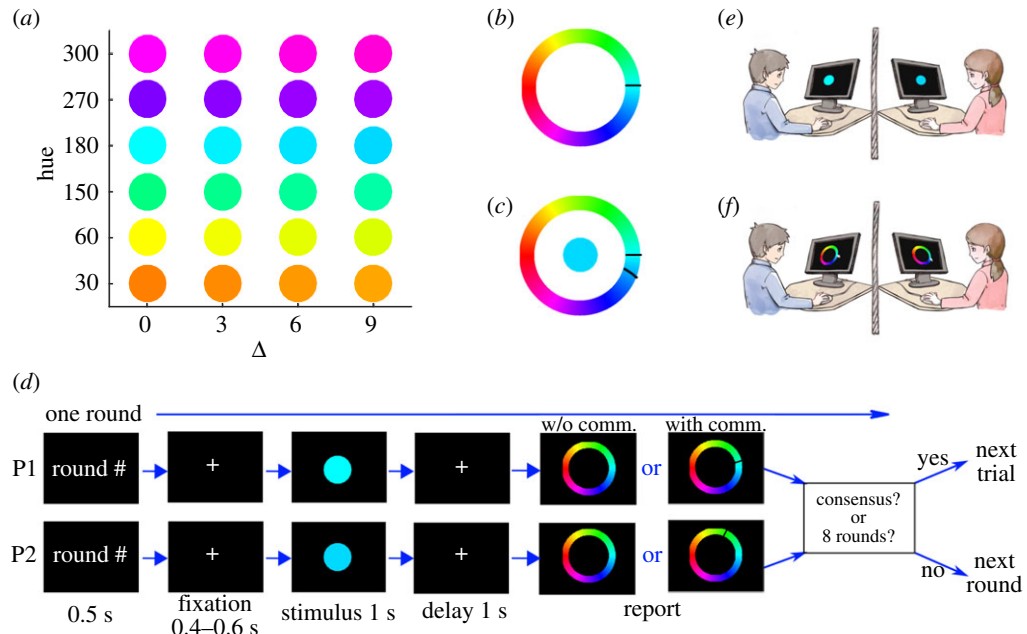

**Figure 1.** Experimental design. (*a*) Coloured discs as stimuli. In one trial, one participant was presented a disc, the colour of which was randomly selected from the first column, and the other participant was presented a disc with the colour selected from the same row with a probability of 0.4, 0.25, 0.25 and 0.1 for a stimulus difference $\Delta = 0°$, $3°$, $6°$ and $9°$, respectively. (*b*) The colour wheel was used to report the estimate, which was randomly rotated across trials. The black bar indicates the report of one's partner in the same dyad from the previous round. (*c*) The trial-end feedback. In the with-communication condition, after the report of the last round of colour estimation, one participant will see the stimulus presented to him/her, his/her own report, and the report of another participant in the dyad. (*d*) The procedure for one round of estimation (see the *text* for details). One trial could be composed of one or more (at most eight) rounds of estimations. Each trial ended once the difference between the reports of two members in one dyad was smaller than 1°; otherwise, the trial moved to the next round. However, one trial ended anyway if both participants had made eight estimations, regardless of whether a consensus had been reached or not. In the with-communication condition, one participant would see the previous-round report of his/her partner as denoted by a black bar on the colour wheel in the report phase. However, in the without-communication condition, one participant would only see a pure colour wheel with no additional signal. (*e*,*f*) A cartoon of the experimental set-up, showing the stimulus presentation phase and report phase respectively. The two participants were isolated to avoid language communication, facial expression communication and gesture communication between them.

Real-time communication between different computers was implemented through Python socket programming using the user datagram protocol (UDP).

## 2.3. Experimental procedure

In this study, we designed two main experimental conditions: with-communication condition and without-communication condition. In the first condition, the members in a dyad were allowed to communicate with each other through a black bar displayed on the colour wheel in the report phase (figure 1*d*). In the second condition, the members in a dyad were not allowed to communicate and we did not show the estimate of the other member in the report phase. Considering that different group members may get different information in the same decision-making scenarios in real life and that the difficulty to reach a consensus may depend on the difference between stimuli presented to participants, we further designed four conditions of stimulus difference wherein colours presented to two members in a dyad might be slightly different. Specifically, we randomly chose the absolute *stimulus difference* $|\Delta| = |s_1 - s_2|$ in one trial to be 0°, 3°, 6° and 9° with probabilities of 0.4, 0.25, 0.25 and 0.1, respectively (figure 1*a*). One member viewed colour $s_1$ and the other viewed colour $s_2$. The colour difference $\Delta$ is with equivalent probabilities for positive and negative values. In each condition of with-communication and without-communication, participants accomplished about 200 trials in total, and trials with different $\Delta$s were interleaved randomly. In each trial, participants were required to estimate and report the colour of a disc as precisely as possible in a round-by-round fashion until two members of a dyad reached a consensus or finished eight rounds of estimation.

In another word, a trial ended immediately when two members had made eight estimations, regardless of whether a consensus had been reached or not. In addition, ten randomly chosen dyads received the trial-end feedback information at the end of a trial in the with-communication condition (figure 1c).

In the with-communication condition, a round started with presenting the round number and then a fixation marker on the screen for a brief period. Then a colour disc (see the examples in figure 1a) was presented for 1 s and participants were requested to report their estimates by clicking on a colour wheel (figure 1d–f). Once the difference between the two participants' reports was smaller than a threshold (1° on the colour wheel), the dyad reached a consensus and the trial ended. Otherwise, the trial moved to the next round. For each round in one trial, each participant was presented the same stimulus as in the first round. One trial would be terminated immediately after eight rounds of reports, regardless of whether a consensus has been reached or not. In the report phase of one round, each participant would see a black bar on the colour wheel (figure 1b) indicating the estimate of his/her partner in the previous round (figure 1d), through which two members in a dyad communicated their estimates. That is, in the second and subsequent rounds in one trial, each participant would see the same stimulus as in the first round and also be notified of his/her partner's previous-round estimate, indicating that one could adjust his/her estimate due to social influence in the with-communication condition.

In the without-communication condition, all the settings were the same as those in the with-communication condition with one exception. We did not show the previous estimate of one participant to another in the report phase. That is, participants would only see a colour wheel with no black bar in the report phase, implying that participants could not communicate with each other.

## 2.4. Data analyses

The experimental results were analysed using Matlab 2015b. First, we excluded trials with mis-operation such as clicking on the background or when the report of either participant deviated from the stimulus by more than 72°. We calculated the circular difference (note that 0° and 360° represent the same colour here) between the report and stimulus, $r - s$, which is called report in the rest of the paper without specific statement. Then we obtained the report distribution of each round for both participants, of which the most interesting ones were those of the first round and the last round (see electronic supplementary material, figure S1 for examples). In the without-communication condition, for trials with $\Delta = 0°$ and no consensus formed, the performance (measured by the mean and standard error of means (s.e.m.) of the report distribution) of a participant did not show consistent improvement as the task proceeded (electronic supplementary material, figure S2). Therefore, it is reasonable to apply the first-round estimates as approximations for the participants' individual estimates, which were not influenced by communication. Further, we found that the bias of the participants' reports for both the first round and the last round were roughly independent of the absolute colours of the stimuli (electronic supplementary material, figure S3). Therefore, in this study we focused on analysing the report distributions, ignoring effects of the absolute colours of the stimuli.

We only took trials with consensus and $\Delta = 0°$ into consideration when we analysed the effects of communication on the quality of the consensus. Once the difference between reports of two participants was smaller than 1°, the consensus was reached and calculated as the mean of reports by the dyad in the last round. Since there is no correct answer for trials with $\Delta \neq 0°$, we did not measure the performance qualities of these trials. Trials with no consensus were also excluded from the performance quality analyses. However, all trials were included in the analyses of speed of consensus formation. Moreover, we assembled all the dyads together to obtain the *aggregation across dyads* to analyse the general properties of the dyadic colour estimation tasks. Note that the significance level of statistical tests is 0.05 if not specified.

# 3. Results

## 3.1. Speed of consensus formation

We first investigated the effects of communication on the speed of consensus formation, i.e. the number of rounds needed to reach a consensus. In the first round where the participants did not communicate with each other, they would only report what they perceived, which followed a distribution around the presented colour with a variation (see electronic supplementary material, figure S1 for examples). From the second round on, if the participants were allowed to communicate with each other, they might adjust their own estimates after seeing their partners' reports. In the with-communication

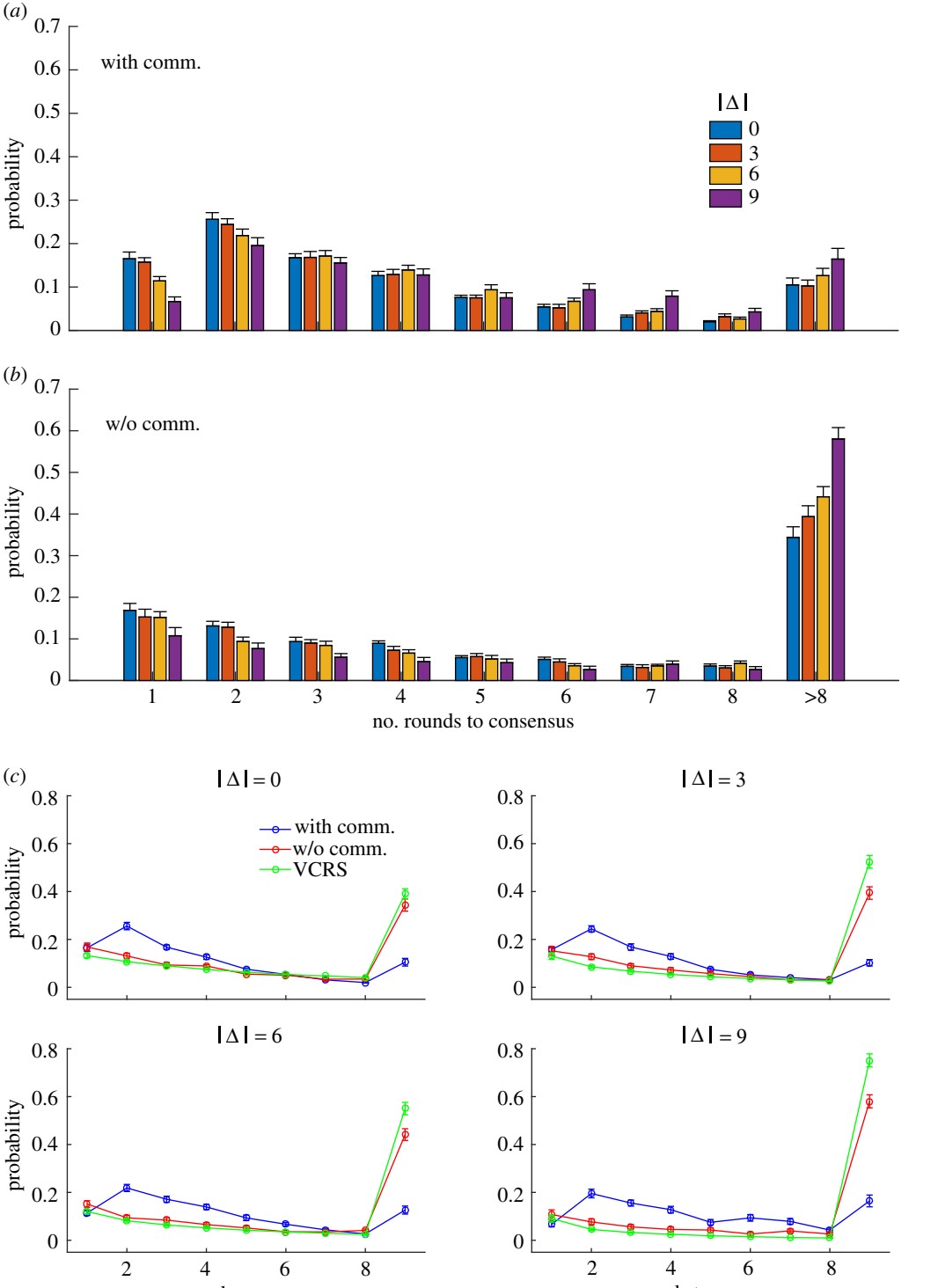

**Figure 2.** Consensus probability as a function of the number of rounds of estimations given a difference between the colours presented to two participants. Different data points represent mean values across all dyads in corresponding conditions. Error bars indicate standard error of means (s.e.m.). (*a*) Participants could communicate with each other. (*b*) Participants were not allowed to communicate with each other. (*c*) Comparisons of the consensus probability among experiments with/without communication and the VCRS model.

condition, the probability of reaching a consensus peaked at the second round of estimation, i.e. the probability initially increased and then decreased approximately with the number of rounds of reports (figure 2*a,c*). Figure 2*a,b* shows the statistical results on consensus formation speed, wherein different data points represent mean values across all dyads in corresponding conditions and error bars indicate s.e.m. In the without-communication condition, the two participants' reports reached a

consensus by chance, guided only by the stimulus. The results showed that the probability of reaching a consensus in the without-communication condition roughly decreased with the number of rounds (figure 2b,c), and these probabilities can be approximately predicted by a model which built virtual consensus by random sampling (VCRS) (figure 2c) (see Supplementary Methods in the electronic supplementary material for details).

The results that two participants reached a consensus within eight rounds of estimations were submitted to a two-factor mixed-design ANOVA to test the effects of communication and stimulus difference. The results showed that the main effect of communication was significant, $F_{1,46} = 151.108$, $p < 0.001$, $\eta_p^2 = 0.767$, which indicated that the probability that participants reached a consensus within eight rounds of estimations in the with-communication condition is significantly higher than that in the without-communication condition. The main effect of stimulus difference was also significant, $F_{3,138} = 36.566$, $p < 0.001$, $\eta_p^2 = 0.443$. The interaction between communication and stimulus difference was significant too, $F_{3,138} = 11.957$, $p < 0.001$, $\eta_p^2 = 0.206$. Simple effect analysis showed that the speed of consensus in the without-communication condition was significantly different among four stimulus difference conditions ($|\Delta| = 0°, 3°, 6°$ and $9°$), $F_{3,44} = 34.757$, $p < 0.001$, $\eta_p^2 = 0.703$. The speed of consensus in the with-communication condition was not significantly different among four stimulus difference conditions ($|\Delta| = 0°, 3°, 6°$ and $9°$), $F_{3,44} = 2.081$, $p = 0.116$, $\eta_p^2 = 0.124$. In short, these results showed that communication between participants significantly speeds up the consensus formation in the dyadic colour estimation task, and the stimulus difference does not significantly affect the speed of consensus formation with communication but severely slows down the speed of the consensus without communication.

## 3.2. Dynamics of consensus formation

We then investigated the dynamic process of consensus formation in this study. We found that participants changed their reports round by round and that their reports gradually approached each other on the way to reaching the final consensus (see figure 3a for examples). If the change between two consecutive reports was large enough, the connecting line between the two consecutive reports of one participant might intersect with that of another participant. This intersection behaviour can be quantified by the ratio of the number of intersections that actually happened over the total number of possible intersections (ratio of intersections, RoI) in one trial or a pool of trials,

$$\mathrm{RoI}(|\Delta|) = \frac{\sum_{R=3}^{8} \sum_{r=0}^{R-2} rN_r}{\sum_{R=3}^{8} \sum_{r=0}^{R-2} (R-2)N_r},$$

where $|\Delta|$ is the absolute stimulus difference between two participants, $R$ is the number of rounds in one trial ($R \in [3, 8]$ means only trials with more than two rounds were considered in this analysis), $r$ is the number of intersections that happened in one trial and $N_r$ is the number of trials in which $r$ intersections happened.

In the without-communication condition, the change of estimate originated from the variation in participants' perception. However, in the with-communication condition, the variation in perception and the influence of the partner's report jointly led to a change in report. We found that the intersections in trials with communication were more frequent than those without communication. Thus RoI in the with-communication condition was, in general, much greater than that in the without-communication condition (figure 3a,b).

We observed that the reports of some dyads had fewer intersections (see the exemplary trials shown in the left panels of figure 3a) and thus a smaller RoI (the first panel in figure 3b), while the reports of other dyads had more intersections (see the exemplary trials shown in the right panels of figure 3a) and thus a larger RoI (the second panel in figure 3b). The RoIs of all dyads given $|\Delta| = 0°, 3°, 6°, 9°$ shown in figure 3c indicates that RoI is dyad-dependent.

We further calculated the adjustment ratio between two consecutive reports,

$$\alpha_i = \frac{r_1(n) - r_1(n-1)}{r_2(n-1) - r_1(n-1)}$$

for one participant, and

$$\beta_i = \frac{r_2(n) - r_2(n-1)}{r_1(n-1) - r_2(n-1)}$$

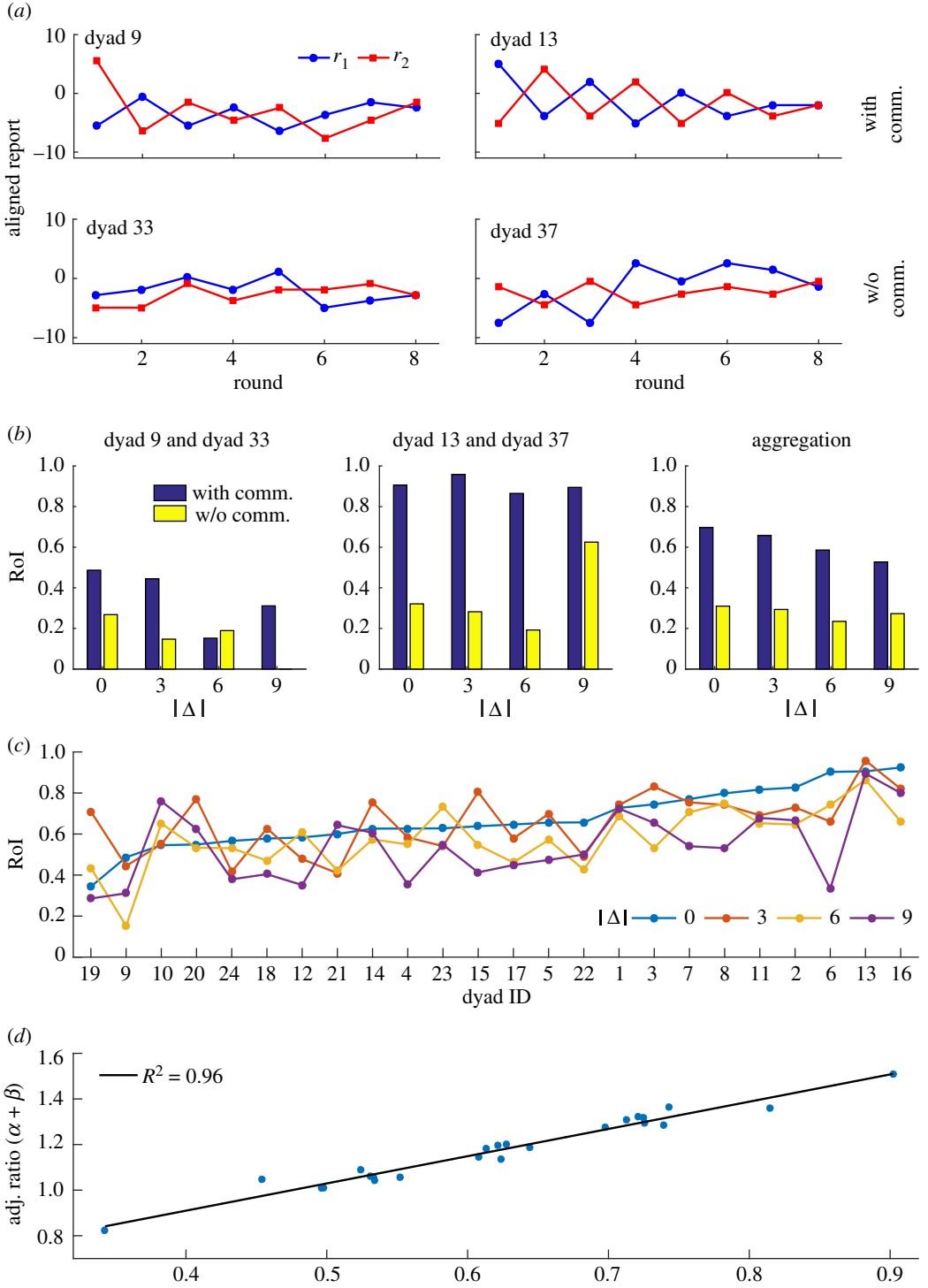

**Figure 3.** Dynamics of consensus formation. (*a*) Examples of the consensus formation process with communication (top) or without communication (bottom). (*b*) Ratio of intersections (RoIs) of exemplary dyads or the aggregation across dyads given different stimulus differences in |Δ|s with communication (blue bar) or without communication (yellow bar). (*c*) RoIs for each dyad given different |Δ|s in the with-communication condition. (*d*) Linear correlation between the sum of the average adjustment ratios and RoI in the with-communication condition.

for another participant, where $r_i(n)$ represents the estimate of the $n$th round of estimation by participant $i$. We found that the sum of the average adjustment ratios (average across rounds and trials) for a dyad $\alpha + \beta$ is linearly correlated with the RoI of that dyad ($R^2 = 0.96$) (figure 3*d*), further confirming that members of a dyad with a higher RoI are more willing to change their minds while members of a dyad with a smaller RoI are reluctant to change.

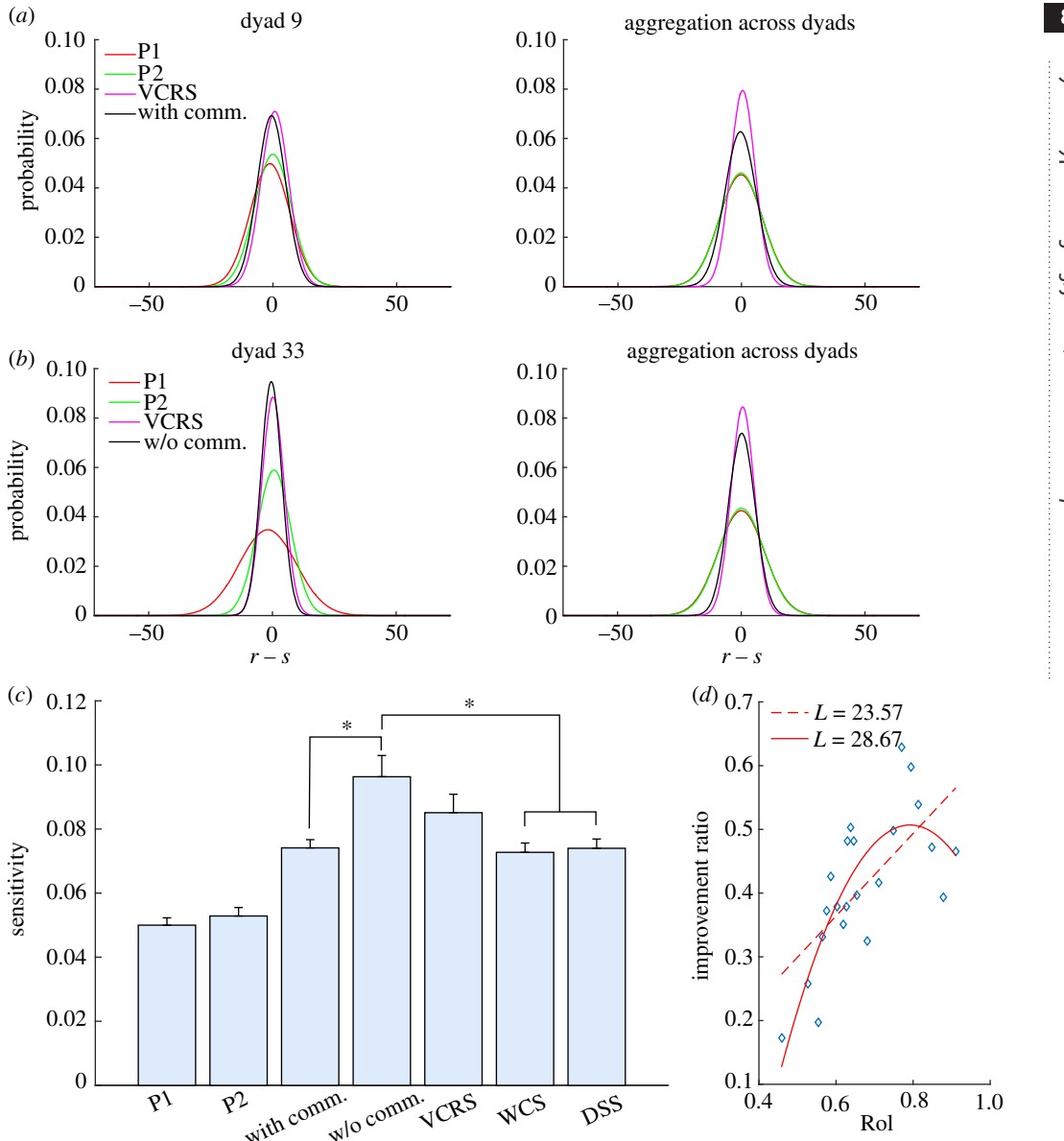

**Figure 4.** Comparison of performances when $\Delta = 0°$. Fitting of the report error distributions for exemplar single participants and dyads (*a*) with or (*b*) without communication, and the corresponding data of the VCRS model. (*c*) Comparison of sensitivities among single participants, dyads with or without communication, and three models including VCRS, WCS and DSS. Asterisks represent significance. Error bars indicate s.e.m. (*d*) Compared to a linear function, a quadratic function better interprets the relationship between the improvement ratio of sensitivity and the ratio of intersections (RoI). *L* means log-likelihood.

## 3.3. Quality of consensus decision

We only investigated the quality of consensus of trials with $\Delta = 0°$ as explained in the *Data analyses* section. To simplify the analysis, we subtracted the value of stimulus from the original reports of participant: $r - s$. Then we obtained the distributions of the reports of participant in the first round and that of consensus reports (see electronic supplementary material, figure S1 for examples). We fitted these distributions using Gaussian function (figure 4*a*,*b*)

$$f(x) = \frac{1}{\sqrt{2\pi}\sigma} \, e^{-\frac{(x-\mu)^2}{2\sigma^2}},$$

where $\mu$ is the bias of the reports and $x = r - s$. We found that $\mu$ is not significantly different from zero for the distribution of the single participant's reports or that of dyadic consensus reports with or without communication (see electronic supplementary material, figure S4 and table S1 for details), indicating

that the single participant's estimates and the dyadic consensus estimates are unbiased. Therefore, we only need to use $\sigma$ to evaluate the quality of estimates. The estimation is more precise and thus the performance is better, if $\sigma$ is smaller. In this study, we described the performance using sensitivity: $s = 1/(\sqrt{2\pi}\sigma)$, which is the same as the definition in [8] and a higher sensitivity of estimates implies a better performance.

It is easy to see that the distribution of consensus estimates with or without communication is apparently narrower than the distribution of the reports from a single participant in one dyad or the aggregation across dyads (figure 4a,b). Consequently, the sensitivity of consensus estimates with communication is significantly larger than that of a single participant's estimates (figure 4c and electronic supplementary material, table S2). These results indicate that the performance of consensus estimates with communication is better than that of individual estimates, which is consistent with the previous studies [8,15]. Note that our experiment cannot discriminate the WCS model with $s_{\mathrm{dyad}}^{\mathrm{WCS}} = (s_1 + s_2)/\sqrt{2}$ from the DSS model with $s_{\mathrm{dyad}}^{\mathrm{DSS}} = \sqrt{s_1^2 + s_2^2}$ (see Supplementary Methods in the electronic supplementary material for details).

To our surprise, we found that the distribution of consensus estimates without communication (figure 4b) is apparently narrower than that of consensus estimates with communication (figure 4a) and the sensitivity of consensus estimates without communication is significantly larger than that of consensus estimates with communication (figure 4c and electronic supplementary material, table S2), which indicates that the performance of consensus estimates without communication is better than the performance of consensus estimates with communication. This phenomenon can be approximately reproduced by the VCRS model (see Supplementary Methods in the electronic supplementary material for details). The distribution produced by the VCRS model is narrower than that of consensus estimates with communication (figure 4a) and is almost the same width as that of consensus estimates without communication (figure 4b). These results imply that dyadic consensus can improve the estimation but the communication between participants in a dyad may impair the improvement of dyadic consensus.

The dyadic consensus improves the performance of estimation by increasing the sensitivity. The improvement ratio of one dyad can be thought as the average of the improvement ratio of participants in the dyad

$$\mathrm{IR} = \left( \frac{s_{\mathrm{dyad}} - s_1}{s_1} + \frac{s_{\mathrm{dyad}} - s_2}{s_2} \right)/2 \times 100\%.$$

We plotted this improvement ratio of one dyad over the corresponding RoI (figure 4d) in which the data points have been smoothed using a sliding window of three points and a sliding step of one point (see electronic supplementary material, figure S5 for the original data). We found that it is better to use a quadratic function ($L = 28.672$) than a linear function ($L = 23.57$) to describe the relationship between the improvement ratio and RoI, where $L$ is the log-likelihood value

$$L = \sum_{i=1}^{n} \log p(\mathrm{RoI}_i, \mathrm{IR}_i)$$

and

$$p(\mathrm{RoI}, \mathrm{IR}) = \frac{1}{\sqrt{2\pi}\sigma}\, \mathrm{e}^{-\frac{(\mathrm{IR} - a\mathrm{RoI}^2 - b\mathrm{RoI} - c)^2}{2\sigma^2}}.$$

A further likelihood ratio test using the *lratiotest* function in Matlab showed that a quadratic function is better to interpret the data than a linear function ($\chi^2(1) = 10.189$, $p = 0.001$). This result suggests that dyads with a moderate RoI have larger improvement of the quality of the consensus estimates, while dyads with an RoI that was either too small or too large have a smaller improvement of the quality of the consensus estimates.

# 4. Discussion

Group decision-making is ubiquitous in human society and animal world, and is very important for their prosperity and development [23–28]. Communication between group members plays a crucial role in group decision-making. However, previous studies demonstrated that whether communication is beneficial or detrimental actually depends on the scenarios [1,3–6]. Recently, Simoiu *et al.* tested the effects of different types of social influence on crowd performance, during a large online experiment

to systematically evaluate crowd performance on 1000 questions across 50 topical domains [29]. And they found that communication can lead to decreasing crowd performance due to herding effect in some instances, while increasing crowd performance in other instances. However, the dynamic process of group decision-making has not yet been clarified and how communication plays its role in the process is still unclear.

In this study, we designed an experimental paradigm to examine the role of communication in the dynamic process of consensus formation and the quality of consensus estimates, in which the communicated information was controllable and the communication was round by round. Due to the simplicity and controllability, our paradigm can possibly be extended to explore the cognitive mechanisms and neural substrates underlying dyadic decision-making or even large-scale group decision-making (see an example in Suzuki *et al.*'s work [30]).

In the dyadic colour estimation experiment, we varied the difference between stimuli and controlled the communicated information between participants, which helped us to understand the role of communication in making group consensus decision. In previous studies, it is not easy to make sure how many rounds of communication happened and how much information has been communicated between participants [8–10,12]. However, in this study, we can easily compare the speed of the consensus formation due to the round-by-round fashion design. We showed that communication of estimates between participants speeds up the consensus formation, as evidenced by the observation that the probability of reaching a consensus within eight rounds of estimations with communication is significantly larger than that without communication (figure 2). We presented different colours to members in dyad to explore how consensus was reached when different group members possessed different information. Our results indicate that the speed of consensus is not affected by the difference between stimuli if the members can communicate their estimates. However, the difference between stimuli severely slows down the speed of consensus when communication is not allowed. Therefore, communication is crucial to the speed of consensus formation in our daily life because group members often have different opinions based on different information they received.

After a detailed look at the dynamic process of the consensus formation in our experiment, we found that there are different communicating patterns. Some dyads have larger RoIs, while others have smaller RoIs (figure 3). If the participants in a dyad change their minds more frequently after viewing the report of their partner, then the dyad may have a large RoI. Otherwise, the dyad has a small RoI. Our further investigation about the improvement of sensitivity shows that the improvement of sensitivity of dyads with moderate RoI is larger than that of dyads with too small or too large RoI (figure 4*d*). This observation indicates that being over-reliant on one's peer or on one's own judgement, could lead to detrimental effects on the consensus. Further work should be designed to test the participants' level of willingness to embrace the choice of others, perhaps with a rating questionnaire.

Surprisingly, we found that consensus without communication outperforms consensus with communication in a statistical sense. Intuitively, in the without-communication condition, a poor consensus requires that both two participants in a dyad make a similarly deviated poor estimate independently, thus reducing the probability of a poor consensus. While in the with-communication condition, two participants affect each other's estimate, thus leading to a higher probability of reaching a poor consensus. Please refer to the Supplementary Methods in the electronic supplementary material for a probabilitic analysis, which mathematically illustrates the mechanism underlying this observation. However, a previous study showed that sensitivity of dyadic decisions without oral communication cannot exceed that of the more sensitive participant [8]. The discrepancy can be explained by the experimental design. In their experimental set-up, the dyadic decision was randomly chosen from one of the two if the participants disagreed, which introduced some randomness to the collective choice and led to the result that the dyad sensitivity cannot exceed that of the more sensitive participant. Participants in our experiment can continue to report their estimates before a consensus formed within eight rounds of estimations, thus the consensus requirement forced the participants to simultaneously make a good or poor choice, decreasing the probability of reaching a poor consensus. Furthermore, the sensitivity of participants can be manipulated by introducing noise to their stimuli without having told them in the previous study [8], but we have not manipulated the sensitivity of the participants, which results in that our experiment could not discriminate the WCS model from the DSS model while their experiment could.

In summary, we showed that dyadic consensus estimate is better than that of each participant in a dyad (figure 4*a*,*b*), which has also been observed in previous studies [8–12]. Besides, our results also showed that communication speeds up consensus formation (figure 2*a*,*b*), but consensus without communication outperforms that with communication (figure 4*a*–*c*). Although these results were

obtained through a perceptual dyadic decision-making task where all information of the stimuli was provided to each participant, they could provide insights into non-perceptual group decision-making scenarios. First, speed–accuracy trade-off is often observed in group or individual decision-making and possible mechanisms have been proposed [31–33]. In this study, we also observed a similar phenomenon, namely, a consensus without communication takes longer but has a higher precision, while a consensus with communication takes a shorter time but has a lower precision. Second, for a non-perceptual group decision-making task where all information has been provided to members of the group, which is similar to our experiment, a better group consensus can be reached after multiple rounds of independent decision of each member. Third, in some non-perceptual group decision-making scenarios, information is incomplete and communication can provide additional information. The members possess more information or more resources usually have higher influence than others when making group decisions. The problem is how to effectively allocate a proper weight to each member in this type of group decision-making. Obviously, this is beyond our experiment, but our study may provide some insights into this problem. For example, we can split the decision process into two stages. In the first stage, group members anonymously communicate with each other to dig out as much information as possible. In the second stage, multiple rounds of independent decisions may help form a better consensus as in our experiment. Of course, this problem deserves more detailed investigations in future.

Ethics. Every participant gave written consent prior to the experiment and received payments after the experiment. This study was approved by the Institutional Review Board and Ethics Committee of Human Participant Protection under IRB no. 201905160043, School of Psychology at Beijing Normal University, Beijing, China.

Data accessibility. Experimental data are available in the electronic supplementary material.

Authors' contributions. L.Y., C.W. and D.-H.W. designed and performed the study. L.Y. and D.-H.W. analysed the data. L.Y., C.W., S.W. and D.-H.W. wrote the manuscript. All authors gave final approval for publication.

Competing interests. We declare we have no competing interests.

Funding. This work was supported by National Natural Science Foundation of China (NSFC) under grant no. 31671077 (D.-H.W.) and grant no. 31900751 (C.W.).

Acknowledgements. We are grateful to Huanyuan Zhou, Ran Liu and Yu Dong, who were graduate students in Beijing Normal University, for their help in collecting data. We also thank Dr KongFatt Wong-lin at Ulster University for his helpful comments.

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
