## [Reviewer comments · Royal Society Open Science]

Review History

Decision letter (RSOS-190903.R0)

28-May-2019

Dear Dr Wang:

Manuscript ID RSOS-190903 entitled "Communication speeds up but impairs the consensus decision in dyadic color estimation task" which you submitted to Royal Society Open Science, has been evaluated.

I must decline the manuscript for publication in Royal Society Open Science at this time. However, a new manuscript may be submitted which takes into consideration the comments below.

Please note that resubmitting your manuscript does not guarantee eventual acceptance, and that your resubmission will be subject to review by reviewer(s) before a decision is rendered.

You will be unable to make your revisions on the originally submitted version of your manuscript. Instead, revise your manuscript using a word processing program and save it on your computer.

Once you have revised your manuscript, go to <https://mc.manuscriptcentral.com/rsos> and login

to your Author Center. Click on "Manuscripts with Decisions," and then click on "Create a Resubmission" located next to the manuscript number. Then, follow the steps for resubmitting your manuscript.

You may also click the below link to start the resubmission process (or continue the process if you have already started your resubmission) for your manuscript. If you use the below link you will not be required to login to ScholarOne Manuscripts.

*** PLEASE NOTE: This is a two-step process. After clicking on the link, you will be directed to a webpage to confirm. ***

https://mc.manuscriptcentral.com/rsos?URL_MASK=a32990fe911e44bea9a615045ce340aa

Because we are trying to facilitate timely publication of manuscripts submitted to Royal Society Open Science, your resubmitted manuscript should be submitted by 25-Nov-2019. If you are unable to submit by this date please contact the Editorial Office for options.

I look forward to a resubmission.

Sincerely,
Andrew Dunn
Senior Publishing Editor
Royal Society Open Science Editorial Office
openscience@royalsociety.org

Editor comments:

The sample size seems terribly low. I would be happy to consider this for a review with a larger sample size and a power calculation attached.

Author's Response to Decision Letter for (RSOS-190903.R0)

See Appendix A.

RSOS-191974.R0

Review form: Reviewer 1

Is the manuscript scientifically sound in its present form?

No

Are the interpretations and conclusions justified by the results?

No

Is the language acceptable?

Yes

Do you have any ethical concerns with this paper?

No

Have you any concerns about statistical analyses in this paper?

Yes

Recommendation?

Major revision is needed (please make suggestions in comments)

Comments to the Author(s)

Yu et al. investigated effects of communication on the dyadic perceptual decision-making in a simple color-estimation task. In the "with-communication" condition participants were able to observe the partner's previous choice, while not in the "no-communication" condition. They found that communication reduced the number of rounds required for consensus formation, but impaired the quality of consensus. I believe the research topic is interesting and will be of wide interest to many researchers. However, I have a couple of concerns about the experimental design, the data analyses, and the interpretations of the data.

According to the authors' response to the editor, they recruited additional participants and there are 48 dyads in the resubmitted version. Did the authors recruit 48 new additional participants? Or, is the total number of participants (the original + the additional) 48? If the former was the case, they need to show the results in the original as well as the additional participants. If the latter is the case, they need to clarify when and how they actually conducted the power analysis (is it a priori?)

"To better determine the effects of communication, we introduced possible differences between the colors of the discs that the two members of a dyad would see in a trial". I don't understand why and how the variable difference between the colors of the discs contributes to better understanding of the effects of communication. Please justify.

"One trial would be terminated after eight rounds of reports, regardless of the consensus of the two participants' reports". How was the maximum number of the rounds in each trial exactly determined? The authors need to provide detailed description.

The authors should provide the explanation about reward payment to the participants. For example, what happens if the two participants could not reach a consensus?

"Communication between participants significantly speeds up consensus formation". Which statistical tests support the conclusion? I could not find the corresponding description.

I don't understand the definition and the interpretation of the psychometric function. First, what does the horizontal axis in Fig. 4ab indicate? How is the curve derived? "The quality of estimation can be described by the psychometric function $P(c)$, which is the probability that a report error is smaller than c . Thus, the steeper the psychometric function, the more sensitive it is". Is it true? If $P(c)$ denotes the probability that a report error is smaller than c . $P(c)$ close to one indicates the sensitive perception. Why the slope does matter? Maybe I misunderstand something; but it would be great if the authors could clarify these points.

How did the authors estimate sensitivity of a single participant? Using the data at the first round is not fair, as the sensitivity should be improved merely by the repetitive exposure to the same stimulus (i.e., even in individual perceptual decision-making, the sensitivity at the first round should be worse than that at the second and later rounds).

In the Bayesian integration models, how participants estimate their partner's confidence-level? Is it possible in this experimental paradigm (c.f., Bahrami et al., 2010)? The authors should provide the comprehensive descriptions about the models (not only for Bayesian models but also all of the other models).

The experimental task differs from other tasks used in the previous studies (e.g., Bahrami et al., 2010) in that pairs of the participants communicated with each other through observing their partner's previous choice. Does the difference in the experimental task accounts for the discrepancy in the findings between the present and the past studies.

Review form: Reviewer 2

Is the manuscript scientifically sound in its present form?

Yes

Are the interpretations and conclusions justified by the results?

No

Is the language acceptable?

No

Do you have any ethical concerns with this paper?

No

Have you any concerns about statistical analyses in this paper?

Yes

Recommendation?

Major revision is needed (please make suggestions in comments)

Comments to the Author(s)

Although, overall I find the paper interesting I have several concerns. Specifically, they are:

Introduction.

There are some important citations missing from the literature review. In particular, it would be nice to put the paper in the context of studies by Miller & Steyvers (2011); and Voinov, Sebanz, & Knoblich (2017) who worked in similar collective decisions in iterative interactions.

Methods

Lines 35 - 38 are redundant and should be omitted.

Line 57: the toolbox is titled "Psychtoolbox" and a proper reference to its website is missing.

Figure 1c caption: "Note that this trial-end feedback might be different for the two members due to the different stimuli they received". I didn't understand how two participants could receive different feedback. More detailed of the procedure in the text would help probably.

Line 21 - 22: This needs to follow immediately after the procedure of the "with-communication" condition to highlight the differences.

Data Analysis

Line 58: Instead of multiple t-tests, the authors should run two-way ANOVA with Communication and Delta as factors. Also, t-tests in Supplementary Table 1 should be reported with p-values corrected for multiple comparisons.

Lines 9 - 18: My suggestion is to move the description of the model to the Supplementary Information, together with specifications of the WCS and DSS models (which are currently only mentioned briefly). Regarding the VCRS model, if I understood it correctly, it sampled from

discrete distributions built from individual independent responses of the the dyad members. Wouldn't a proper model of the process sample from continuous distributions (viz., Gaussian), estimated from individual response data? After all, participants' responses are made on a continuous scale, which makes a consensus by chance so difficult to emerge. Is this the reason why the VCRS model underestimates the rate of trials which never reached a consensus in the without-communication condition?

Figures 4a and 4b "Dyad collection" are missing P1 curves.

Figure 4d. I am not convinced that the inverse U function is the best fit to the data, and its interpretations are based on a rather weak evidence. If the authors want to make a claim that on the relationship between RoI and Improvement Ratio is not just linear, but that a quadratic component of the model explains more variation in the data, they need to make a proper model comparison of the log-likelihood values.

Lines 57 - 62. Participants' performance in the no-communication condition, as well as predictions from the VCRS model, was more accurate than the optimal Bayesian model. How can this be? My understanding is that the DSS and WCS models take as an input only one observation (that is, from the first round of a trial), while the VCRS model makes use of multiple observations through consecutive rounds. In other words, the algorithm collects multiple samples which results in a more reliable integration. All this needs to be spelled out in the text.

A general question on the "estimations quality" data. How was data aggregated when consensus was not reached? Were two judgments from the last round averaged or was an average accuracy of the two judgments taken? This needs to be specified explicitly.

Discussion

The most important aspect not touched in the discussion, is why dyads performed more accurate in the no - communication condition, and why they outperformed a Bayesian optimal algorithm. The authors provide no psychological interpretation for this interesting finding, and the reader is left to wonder why one mode of interaction turns out to be more effective than another. The exact mechanism underlying the effect needs to be spelled out here.

The paper would benefit from discussing some broader implications of the findings, including non-perceptual decision-making problems. In which situations collective estimations benefit from communication and when is communication detrimental? The conclusions made, e.g. "However, it might be beneficial that group members share information about the problem before making their own decisions to gather as much information as possible" are not related to the results.

The biggest problem with the manuscript, thought, is its language and multiple grammatical and stylistic errors throughout the paper. The authors are advised to have their manuscript proof-read, if possible by a native English speaker, before they submit the final version.

Decision letter (RSOS-191974.R0)

26-Feb-2020

Dear Dr Wang,

The Subject Editor assigned to your paper ("Communication speeds up but impairs the consensus decision in dyadic color estimation task") has now received comments from reviewers. We

would like you to revise your paper in accordance with the referee and Associate Editor suggestions which can be found below (not including confidential reports to the Editor). Please note this decision does not guarantee eventual acceptance.

Please submit a copy of your revised paper before 20-Mar-2020. Please note that the revision deadline will expire at 00.00am on this date. If we do not hear from you within this time then it will be assumed that the paper has been withdrawn. In exceptional circumstances, extensions may be possible if agreed with the Editorial Office in advance. We do not allow multiple rounds of revision so we urge you to make every effort to fully address all of the comments at this stage. If deemed necessary by the Editors, your manuscript will be sent back to one or more of the original reviewers for assessment. If the original reviewers are not available we may invite new reviewers.

When submitting your revised manuscript, you must respond to the comments made by the referees and upload a file "Response to Referees" in "Section 6 - File Upload". Please use this to document how you have responded to each of the comments, and the adjustments you have made. In order to expedite the processing of the revised manuscript, please be as specific as possible in your response.

- Ethics statement

- Data accessibility

<http://datadryad.org/submit?journalID=RSOS&manu=RSOS-191974>

- Competing interests

- Authors' contributions

All submissions, other than those with a single author, must include an Authors' Contributions section which individually lists the specific contribution of each author. The list of Authors

should meet all of the following criteria; 1) substantial contributions to conception and design, or acquisition of data, or analysis and interpretation of data; 2) drafting the article or revising it critically for important intellectual content; and 3) final approval of the version to be published.

- Acknowledgements

- Funding statement

on behalf of Dr Simone Schnall (Associate Editor) and Essi Viding (Subject Editor)
openscience@royalsociety.org

Associate Editor Comments to Author (Dr Simone Schnall):

Comments to the Author:

We've received two reports on your resubmitted manuscript. Please ensure you address all comments raised by the referees within a point-by-point response upon submitting your revision. Particularly, please ensure you further clarify and justify sections within your manuscript, and provide further detailed descriptions.

Reviewer comments to Author:

Reviewer: 1

Comments to the Author(s)

Yu et al. investigated effects of communication on the dyadic perceptual decision-making in a simple color-estimation task. In the "with-communication" condition participants were able to observe the partner's previous choice, while not in the "no-communication" condition. They found that communication reduced the number of rounds required for consensus formation, but impaired the quality of consensus. I believe the research topic is interesting and will be of wide interest to many researchers. However, I have a couple of concerns about the experimental design, the data analyses, and the interpretations of the data.

According to the authors' response to the editor, they recruited additional participants and there are 48 dyads in the resubmitted version. Did the authors recruit 48 new additional participants? Or, is the total number of participants (the original + the additional) 48? If the former was the case, they need to show the results in the original as well as the additional participants. If the latter is the case, they need to clarify when and how they actually conducted the power analysis (is it a priori?)

"To better determine the effects of communication, we introduced possible differences between the colors of the discs that the two members of a dyad would see in a trial". I don't understand why and how the variable difference between the colors of the discs contributes to better understanding of the effects of communication. Please justify.

"One trial would be terminated after eight rounds of reports, regardless of the consensus of the two participants' reports". How was the maximum number of the rounds in each trial exactly determined? The authors need to provide detailed description.

The authors should provide the explanation about reward payment to the participants. For example, what happens if the two participants could not reach a consensus?

"Communication between participants significantly speeds up consensus formation". Which statistical tests support the conclusion? I could not find the corresponding description.

I don't understand the definition and the interpretation of the psychometric function. First, what does the horizontal axis in Fig. 4ab indicate? How is the curve derived? "The quality of estimation can be described by the psychometric function $P(c)$, which is the probability that a report error is smaller than c . Thus, the steeper the psychometric function, the more sensitive it is". Is it true? If $P(c)$ denotes the probability that a report error is smaller than c . $P(c)$ close to one indicates the sensitive perception. Why the slope does matter? Maybe I misunderstand something; but it would be great if the authors could clarify these points.

How did the authors estimate sensitivity of a single participant? Using the data at the first round is not fair, as the sensitivity should be improved merely by the repetitive exposure to the same stimulus (i.e., even in individual perceptual decision-making, the sensitivity at the first round should be worse than that at the second and later rounds).

In the Bayesian integration models, how do participants estimate their partner's confidence-level? Is it possible in this experimental paradigm (c.f., Bahrami et al., 2010)? The authors should provide the comprehensive descriptions about the models (not only for Bayesian models but also all of the other models).

The experimental task differs from other tasks used in the previous studies (e.g., Bahrami et al., 2010) in that pairs of the participants communicated with each other through observing their partner's previous choice. Does the difference in the experimental task account for the discrepancy in the findings between the present and the past studies.

Reviewer: 2

Comments to the Author(s)

Although, overall I find the paper interesting I have several concerns. Specifically, they are:

Introduction.

There are some important citations missing from the literature review. In particular, it would be nice to put the paper in the context of studies by Miller & Steyvers (2011); and Voinov, Sebanz, & Knoblich (2017) who worked in similar collective decisions in iterative interactions.

Methods

Lines 35 - 38 are redundant and should be omitted.

Line 57: the toolbox is titled "Psychtoolbox" and a proper reference to its website is missing.

Figure 1c caption: "Note that this trial-end feedback might be different for the two members due to the different stimuli they received". I didn't understand how two participants could receive different feedback. More detailed of the procedure in the text would help probably.

Line 21 - 22: This needs to follow immediately after the procedure of the "with-communication" condition to highlight the differences.

Data Analysis

Line 58: Instead of multiple t-tests, the authors should run two-way ANOVA with Communication and Delta as factors. Also, t-tests in Supplementary Table 1 should be reported with p-values corrected for multiple comparisons.

Lines 9 - 18: My suggestion is to move the description of the model to the Supplementary Information, together with specifications of the WCS and DSS models (which are currently only mentioned briefly). Regarding the VCRS model, if I understood it correctly, it sampled from discrete distributions built from individual independent responses of the the dyad members. Wouldn't a proper model of the process sample from continuous distributions (viz., Gaussian), estimated from individual response data? After all, participants' responses are made on a continuous scale, which makes a consensus by chance so difficult to emerge. Is this the reason why the VCRS model underestimates the rate of trials which never reached a consensus in the without-communication condition?

Figures 4a and 4b "Dyad collection" are missing P1 curves.

Figure 4d. I am not convinced that the inverse U function is the best fit to the data, and its interpretations are based on a rather weak evidence. If the authors want to make a claim that on the relationship between RoI and Improvement Ratio is not just linear, but that a quadratic component of the model explains more variation in the data, they need to make a proper model comparison of the log-likelihood values.

Lines 57 - 62. Participants' performance in the no-communication condition, as well as predictions from the VCRS model, was more accurate than the optimal Bayesian model. How can this be? My understanding is that the DSS and WCS models take as an input only one observation (that is, from the first round of a trial), while the VCRS model makes use of multiple observations through consecutive rounds. In other words, the algorithm collects multiple samples which results in a more reliable integration. All this needs to be spelled out in the text.

A general question on the "estimations quality" data. How was data aggregated when consensus was not reached? Were two judgments from the last round averaged or was an average accuracy of the two judgments taken? This needs to be specified explicitly.

Discussion

The most important aspect not touched in the discussion, is why dyads performed more accurate in the no - communication condition, and why they outperformed a Bayesian optimal algorithm. The authors provide no psychological interpretation for this interesting finding, and the reader is left to wonder why one mode of interaction turns out to be more effective than another. The exact mechanism underlying the effect needs to be spelled out here.

The paper would benefit from discussing some broader implications of the findings, including

non-perceptual decision-making problems. In which situations collective estimations benefit from communication and when is communication detrimental? The conclusions made, e.g. "However, it might be beneficial that group members share information about the problem before making their own decisions to gather as much information as possible" are not related to the results.

The biggest problem with the manuscript, thought, is its language and multiple grammatical and stylistic errors throughout the paper. The authors are advised to have their manuscript proof-read, if possible by a native English speaker, before they submit the final version.

Author's Response to Decision Letter for (RSOS-191974.R0)

See Appendix B.

RSOS-191974.R1 (Revision)

Review form: Reviewer 1

Is the manuscript scientifically sound in its present form?

No

Are the interpretations and conclusions justified by the results?

No

Is the language acceptable?

Yes

Do you have any ethical concerns with this paper?

No

Have you any concerns about statistical analyses in this paper?

Yes

Recommendation?

Major revision is needed (please make suggestions in comments)

Comments to the Author(s)

I acknowledge the authors' efforts to revise the manuscript. I appreciate it. However, I still have a couple of concerns.

[Major concerns]

My critical concern is about the way of the data analysis. As far as I understand, what the authors actually did is (1) to get the original data; (2) to get the additional data; and (3) to analyze the original and the additional data together. The sample size (the number of participants in the original + that in the additional data) was determined by a power analysis based on the effect size estimated from the original data. This procedure was not consistent with the standard way of

hypothesis testing. In other words, pilot data for a power analysis to determine sample size should be independent from the main data for testing the hypothesis. I believe what the authors should have done is (1) to get the original data; (2) to conduct a power analysis based on the original data in order to determine sample size of additional data; (3) to get the additional data; (3) to analyze the additional data separated from the original data.

If I understand correctly, the present study suggests that people are more likely to reach a consensus in the presence of communication, compared with the case in the absence of communication. On the other hand, if we focus ONLY on the cases in which the pair of participants reached a consensus, the dyadic consensus estimates in the absence of communication outperformed those in the presence of communication. Given that, I believe the claim that consensus estimation without communication generally outperforms that with communication is an overstatement.

[Minor concern]

In Figure 4c, it would be interesting to show the data about the "bias" not only the "sensitivity". While the authors mentioned that the bias was not significantly different from zero, there might be differences between the conditions.

Review form: Reviewer 2

Is the manuscript scientifically sound in its present form?

Yes

Are the interpretations and conclusions justified by the results?

Yes

Is the language acceptable?

No

Do you have any ethical concerns with this paper?

No

Have you any concerns about statistical analyses in this paper?

No

Recommendation?

Accept with minor revision (please list in comments)

Comments to the Author(s)

Most of the issues I raised have been properly addressed in the revised version. Some minor issues would still need to be addressed though(see below):

Lines 50 - 53 reads as if it was an ad, it would be better to remove it.

Line 74 - 75" In each condition of with-communication and without-communication, participants accomplished about 200 trials in total"

I didn't understand why the authors didn't provide the exact number of trials. Was it different for different dyads? If so, why was there variable amount of trials in different conditions for different dyads?

The term "Dyad Collection" is an improper one. Probably, the authors meant "Aggregated across Dyads" or simply "Averaged".

Unnumbered Lines between Line 257 and 258. In general, I find it not a so good idea to fill the Discussion section with technical details unless they are essential to the final conclusions (which is not the case here). The manuscript would benefit if the authors move the formulae from this section either to the Supplementary Information or to the Appendix and rephrase their explanation in a less technical, compressed, way.

I also found that, although the rest of the manuscript has been proof-read and so now it reads nice and neat, the new parts of the Discussion section are still replete with syntax errors. It is very desirable to have another round of proof-reading for this specific part.

Decision letter (RSOS-191974.R1)

Dear Dr Wang:

On behalf of the Editors, I am pleased to inform you that your Manuscript RSOS-191974.R1 entitled "Communication speeds up but impairs the consensus decision in a dyadic color estimation task" has been accepted for publication in Royal Society Open Science subject to minor revision in accordance with the referee suggestions. Please find the referees' comments at the end of this email.

The reviewers and Subject Editor have recommended publication, but also suggest some minor revisions to your manuscript. Therefore, I invite you to respond to the comments and revise your manuscript.

- Ethics statement

- Data accessibility

<http://datadryad.org/submit?journalID=RSOS&manu=RSOS-191974.R1>

- **Competing interests**

- **Authors' contributions**

- **Acknowledgements**

- **Funding statement**

Because the schedule for publication is very tight, it is a condition of publication that you submit the revised version of your manuscript before 21-Jun-2020. Please note that the revision deadline will expire at 00.00am on this date. If you do not think you will be able to meet this date please let me know immediately.

1) A text file of the manuscript (tex, txt, rtf, docx or doc), references, tables (including captions) and figure captions. Do not upload a PDF as your "Main Document".

- 2) A separate electronic file of each figure (EPS or print-quality PDF preferred (either format should be produced directly from original creation package), or original software format)
- 3) Included a 100 word media summary of your paper when requested at submission. Please ensure you have entered correct contact details (email, institution and telephone) in your user account
- 4) Included the raw data to support the claims made in your paper. You can either include your data as electronic supplementary material or upload to a repository and include the relevant doi within your manuscript
- 5) All supplementary materials accompanying an accepted article will be treated as in their final form. Note that the Royal Society will neither edit nor typeset supplementary material and it will be hosted as provided. Please ensure that the supplementary material includes the paper details where possible (authors, article title, journal name).

on behalf of Dr Simone Schnall (Associate Editor) and Essi Viding (Subject Editor)
openscience@royalsociety.org

Associate Editor Comments to Author (Dr Simone Schnall):

Comments to the Author:

As the reviewers note, the manuscript is much improved, and I am therefore happy to accept it pending a minor revision that addresses the remaining points listed by the reviewers. In particular, please note the concern raised by Reviewer 1 of how the new data were combined with the existing data (e.g., discuss the implications of this in a footnote in the paper). Please also ensure that the entire manuscript, incl. new sections in the Discussion, is proof-read to ensure proper English grammar and syntax.

Reviewer comments to Author:
Reviewer: 1

Comments to the Author(s)

I acknowledge the authors' efforts to revise the manuscript. I appreciate it. However, I still have a couple of concerns.

[Major concerns]

My critical concern is about the way of the data analysis. As far as I understand, what the authors actually did is (1) to get the original data; (2) to get the additional data; and (3) to analyze the original and the additional data together. The sample size (the number of participants in the original + that in the additional data) was determined by a power analysis based on the effect size estimated from the original data. This procedure was not consistent with the standard way of hypothesis testing. In other words, pilot data for a power analysis to determine sample size should be independent from the main data for testing the hypothesis. I believe what the authors should have done is (1) to get the original data; (2) to conduct a power analysis based on the original data in order to determine sample size of additional data; (3) to get the additional data; (3) to analyze the additional data separated from the original data.

If I understand correctly, the present study suggests that people are more likely to reach a consensus in the presence of communication, compared with the case in the absence of communication. On the other hand, if we focus ONLY on the cases in which the pair of participants reached a consensus, the dyadic consensus estimates in the absence of communication outperformed those in the presence of communication. Given that, I believe the claim that consensus estimation without communication generally outperforms that with communication is an overstatement.

[Minor concern]

In Figure 4c, it would be interesting to show the data about the "bias" not only the "sensitivity". While the authors mentioned that the bias was not significantly different from zero, there might be differences between the conditions.

Reviewer: 2

Comments to the Author(s)

Most of the issues I raised have been properly addressed in the revised version. Some minor issues would still need to be addressed though(see below):

Lines 50 - 53 reads as if it was an ad, it would be better to remove it.

Line 74 - 75" In each condition of with-communication and without-communication, participants accomplished about 200 trials in total"

I didn't understand why the authors didn't provide the exact number of trials. Was it different for different dyads? If so, why was there variable amount of trials in different conditions for different dyads?

The term "Dyad Collection" is an improper one. Probably, the authors meant "Aggregated across Dyads" or simply "Averaged".

Unnumbered Lines between Line 257 and 258. In general, I find it not a so good idea to fill the Discussion section with technical details unless they are essential to the final conclusions (which is not the case here). The manuscript would benefit if the authors move the formulae from this section either to the Supplementary Information or to the Appendix and rephrase their explanation in a less technical, compressed, way.

I also found that, although the rest of the manuscript has been proof-read and so now it reads nice and neat, the new parts of the Discussion section are still replete with syntax errors. It is very desirable to have another round of proof-reading for this specific part.

Author's Response to Decision Letter for (RSOS-191974.R1)

See Appendix C.

Decision letter (RSOS-191974.R2)

Dear Dr Wang,

It is a pleasure to accept your manuscript entitled "Communication speeds up but impairs the consensus decision in a dyadic color estimation task" in its current form for publication in Royal Society Open Science. The comments of the reviewer(s) who reviewed your manuscript are included at the foot of this letter.

on behalf of Dr Simone Schnall (Associate Editor) and Essi Viding (Subject Editor)
openscience@royalsociety.org

Associate Editor Comments to Author (Dr Simone Schnall):

Comments to the Author:

Dear Mr. Wang, thank you for addressing the reviewers' final comments, and for your patience in carrying out an extensive amount of revision on this manuscript overall. It is my pleasure to now accept it for publication.

Appendix A

Dear Editor,

According to the suggestion of editor to our previous manuscript entitled “Communication speeds up but impairs the consensus decision in dyadic decision making”, we recruited additional participants in the past months and there are 48 dyads in the resubmitted version. Half dyads have participated in tasks in the with-communication condition, and the rest has participated in tasks in the no-communication condition. The sample size was determined a priori from a power analysis (G*Power), which indicated that the number of participants was sufficient to detect a between-subjects effect (Cohen's $d = 0.964$) 90% of the time using a significance level of 0.05.

Now we have a larger sample size and the results remain unchanged as before. Besides, we also some revisions as follows.

- 1) We updated all figures involving results about all dyads considering the new participants. Error bars in all figures represent standard error of means (s.e.m.).
- 2) In the section “*Participants*”, we added the reason why we chose the current sample size.
- 3) We redrew the procedure for one round of estimation (Fig.1 d) to show two different experimental conditions (with-communication versus no-communication) explicitly. Correspondingly, we rephrased the descriptions in the text and in the figure caption. We also rearranged Fig.1 to make it look better.
- 4) We rearranged the structure of the section “*Speed of consensus formation*”, and we deleted some vague descriptions, to make this part more intelligible. Further, we added statistical tests to support our conclusion that communication between participants significantly speeds up consensus formation in this dyadic low-level perceptual decision-making task.
- 5) In Fig.4 d, we applied a moving average to smooth the data points and then fit the smoothed data, which highlights the effects of intersection on the improvement of the final decision. The original data was shown in Supp. Fig.3.

Best regards

DaHui Wang

Professor of School of Systems Science
State Key Laboratory of Cognitive Neuroscience and learning
Beijing Normal University

Appendix B

Dear Editor,

Thanks for the helpful comments and suggestions. We have made substantial revisions according to the comments of reviewers. The major revisions are as follows.

- (1) In the section “Introduction”, we have introduced more background information to better explain why we conducted this study. We have also simplified the redundant descriptions about the experimental results, and presented some hypotheses instead.
- (2) In the section “Participants”, we have added details about the Ethics statement, and the reward payments to the participants. And we also explained how we determined the sample size.
- (3) In the section “Experimental setup”, we corrected the toolbox name “Psychtoolbox-3”, and provided more information for the toolbox.
- (4) We have almost rewritten the section “Experimental procedure”. We provided more details about the experimental design and the process of a trial to make it more clarified. Specifically, we first described the experimental conditions (with/without communication, and 4 absolute stimulus differences), and then explained the process of a trial in the with-communication and without-communication condition respectively. Accordingly, we also updated Fig. 1, specifically Fig. 1 d, and the figure caption.
- (5) In the section “Data analyses”, we explained the reason why we applied the first-round estimates as approximations of the participants’ individual estimates, and the reason why we only took trials with consensus and $\Delta = 0^\circ$ into consideration when we analyzed the effects of communication on performance quality.
- (6) In the section “Speed of consensus formation”, we have added a two-factor mixed-design ANOVA to test the effects of communication and stimulus difference. We have also updated the details of the VCRS model, and moved it to Supplementary Methods. Accordingly, we have updated Fig. 2 c to show the new results.
- (7) In the section “Quality of consensus decision”, we have replaced the psychometric function with the Gaussian function to measure the estimation quality (Figs. 4 a and b). We observed that the estimations were not biased in the single participant’s reports, and dyadic consensus reports with or without communication. Then we applied sensitivity $s = \frac{1}{\sqrt{2\pi}\sigma}$ to measure estimation quality, and the main results stay unchanged as the previous version (Fig. 4 c). Moreover, we applied maximum log likelihood estimation to approximate the parameters of the fitting function of the improvement ratio (IR) over the ratio of intersections (RoI) (Fig. 4 d). We also

calculated the log-likelihood values of a quadratic function and a linear function, and found that the former better interprets the data. Accordingly, we have updated Fig. 4 to show the new results.

- (8) We almost rewrote the section "Discussion", we clarified the mechanism that consensus without communication outperforms consensus with communication. We extend our discussion of the implications of the experimental diagram and the results, including the scenarios where group members possess different information, the broader effects of communication, the non-perceptual decision-making problems, and so on.
- (9) We moved the details of the models (VCRS, WCS, DSS) into Supplementary Materials. We have also presented the statistical results about report biases in the Supplementary Materials. Moreover, we have updated the statistical test results about sensitivities, and report p-values corrected using Bonferroni correction, as suggested by Reviewer 2.

Thanks again for the helpful comments and suggestions.

Please find below the point-by-point reply.

Best wishes,

Yours sincerely,

DaHui Wang

Reviewer comments to Author:

Reviewer: 1

Comments to the Author(s)

Yu et al. investigated effects of communication on the dyadic perceptual decision-making in a simple color-estimation task. In the “with-communication” condition participants were able to observe the partner’s previous choice, while not in the “no-communication” condition. They found that communication reduced the number of rounds required for consensus formation, but impaired the quality of consensus. I believe the research topic is interesting and will be of wide interest to many researchers. However, I have a couple of concerns about the experimental design, the data analyses, and the interpretations of the data.

According to the authors’ response to the editor, they recruited additional participants and there are 48 dyads in the resubmitted version. Did the authors recruit 48 new additional participants? Or, is the total number of participants (the original + the additional) 48? If the former was is case, they need to show the results in the original as well as the additional participants. If the latter is the case, they need to clarify when and how they actually conducted the power analysis (is it a priori)?

Reply: Thanks for the comments. We did not recruit 48 new additional participants. The original and the additional participants were arranged into 48 dyads in total. Thus, 48 is the total number of dyads that have attended this experiment, including the original and additional ones. We conducted the power analysis after we received the editor’s requirement to increase the number of subjects, but before the additional part of the experiment. And we have deleted the misleading term “a priori” in the text.

Here, to make sure that 48 is enough to support the present research, we also describe the changing trend of the intensity of Bayesian factors between two groups of participants under different conditions. Specifically, the following figures respectively show how the Bayesian factors (BF) change with the increase of sample size under the four conditions of stimulus difference through Bayesian sequence analysis. When the sample size reaches 40, the Bayesian factor values under all four conditions are more than 100000 (BF > 10, strong evidence), and convergence begins to appear. Therefore, the current research sample is sufficient.

$BF_{10} = 16400000$
 $BF_{01} = 6.09e-08$

- user
- wide
- ultrawide

$BF_{10} = 9.33e+09$
 $BF_{01} = 1.07e-10$

- user
- wide
- ultrawide

“To better determine the effects of communication, we introduced possible differences between the colors of the discs that the two members of a dyad would see in a trial”. I don’t understand why and how the variable difference between the colors of the discs contributes to better understanding of the effects of communication. Please justify.

Reply: Thank you for the question. There are two reasons why we introduced the difference between stimuli presented to participants. First, in real life, different members in a group may receive different information in a group decision-making scenario due to various reasons. Even group members receive different information, they can often reach a consensus. Second, we hypothesized that it should be more difficult to reach a consensus if we presented stimuli with large difference. Thus, we introduced difference between the colors presented to two members in some trials. We found that participants can reach consensus given different colors, but we cannot measure the quality of the consensus in those trials. We also found the speed of consensus formation will not be significantly affected by difference of colors if participants can communicate with each other. However, the difference will significantly slow down the consensus formation if the communication between participants is not allowed, indicating that communication makes it easy to reach a consensus.

We have rewritten this part in the “Experimental procedure” section to make it clearer (line 67-74) as follows:

Considering that different group members may get different information in the same decision-making scenario in real life and that the difficulty to reach a consensus may depend on the difference between stimuli presented to participants, we further designed four conditions of stimulus difference wherein colors presented to two members in a dyad might be slightly different. Specifically, we randomly chose the absolute stimulus difference $|\Delta| = |s_1 - s_2|$ in one trial to be $0^\circ, 3^\circ, 6^\circ,$ and 9° with probabilities of 0.4, 0.25, 0.25, and 0.1, respectively (Fig. 1 a). One member viewed color s_1 and the other viewed color s_2 . The color difference Δ is with equivalent probabilities for positive and negative values.

We have done statistical analysis on the effects of stimulus difference on the speed of consensus formation. We show these results in the second paragraph in section “Speed of consensus formation” and summarized the results in (line 156-160):

“In short, these results showed that communication between participants significantly speeds up the consensus formation in the dyadic color estimation task, and the stimulus difference does not significantly affect the speed of consensus formation with communication but severely slows down the speed of the consensus without communication.”

“One trial would be terminated after eight rounds of reports, regardless of the consensus of the two participants’ reports”. How was the maximum number of the rounds in each trial exactly determined? The authors need to provide detailed description.

Reply: Thanks for the helpful suggestion. We have added more details about the experimental design in the revised manuscript (line 76-80). We have also updated Fig. 1 and its caption to clarify the experimental design. One trial could be composed of one or more (at most eight) rounds of estimations. 1) Each trial ended once the difference between the reports of two members in one dyad was smaller than the threshold 1° ; otherwise, the trial moved to next round. 2) One trial ended immediately if the dyad has reported their estimates of the eighth round, regardless of whether a consensus had been formed or not.

The authors should provide the explanation about reward payment to the participants. For example, what happens if the two participants could not reach a consensus?

Reply: Thanks for the comments. We have clarified the reward payments to the participants in the revised manuscript (line 35-39) as follows:

Each participant received RMB 60 yuan per hour. To encourage the participants to reach a consensus as accurately as possible, we provided RMB 100 yuan as a bonus to the top-ten dyads in each condition with regard to the accuracy of their consensus from trials in which they received the same stimulus. Whether reaching a consensus or not in one trial had no influence on their payments. This task had no requirement on response speed.

“Communication between participants significantly speeds up consensus formation”. Which statistical tests support the conclusion? I could not find the corresponding description.

Reply: Sorry for the confusion. In the revised manuscript, we applied ANOVA to analyze the data, as suggested by another reviewer. The new results were placed in section “Speed of consensus formation” (line 144-149) as follows.

The results that two participants reached a consensus within eight rounds of estimations were submitted to a two-factor mixed-design ANOVA to test the effects of communication and stimulus difference. The results showed that the main effect of communication was significant, $F(1, 46) = 151.108$, $p < 0.001$, $\eta_p^2 = 0.767$, which indicated that the probability that participants reached a consensus within eight rounds of reports in the with-communication condition is significantly higher than that in the without-communication condition.

I don't understand the definition and the interpretation of the psychometric function. First, what does the horizontal axis in Fig. 4ab indicates? How is the curve derived? “The quality of estimation can be described by the psychometric function $P(c)$, which is the probability that a report error is smaller than c . Thus, the steeper the psychometric function, the more sensitive it is”. Is it true? If $P(c)$ denotes the probability that a report error is smaller than c . $P(c)$ close to one indicates the sensitive perception. Why the slope does matter? Maybe I misunderstand something; but it would be great if the authors could clarify these points.

Reply: Sorry for the confusion. To make it understood, we adopted the probability density function (PDF) of participants' estimates instead of the psychometric function in the revised manuscript and rewritten the whole section “Quality of consensus decision”.

In Bahrami et al. 2010, the authors obtained the psychometric function by calculating the proportion of trials in which the oddball was reported in the second interval is plotted against the contrast difference at the oddball location. They fitted the psychometric curve using the cumulative Gaussian function $P(\Delta c) = H\left(\frac{\Delta c + b}{\sigma}\right)$, where Δc was the contrast difference, $P(\Delta c)$ corresponded to the probability of saying that the second interval had

the higher contrast, and $H(z) = \int_{-\infty}^z \frac{dt}{\sqrt{2\pi}} \exp[-t^2/2]$. This slope of the psychometric function at zero contrast reflects the sensitivity of the contrast detectability. For example, if the psychometric function is a step function at zero contrast, the participant can detect the oddball even there is only a very tiny nonzero contrast. If the psychometric function is a horizontal line, the participant cannot detect the oddball even the contrast is large and the participant just randomly report. This is the reason why we claimed “the steeper the psychometric function, the more sensitive it is”. Further, if we request the participant to report the contrast difference, we can obtain a probability density function of the report, which should be the derivate of the psychometric function. If this PDF is a Gaussian function, the standard deviation is inverse to the slope of the psychometric function at zero contrast difference. Thus, we have the statements in the original version.

In the revised version of the manuscript, we used probability density function which was derived from the proportion of the binned reports (Figs. 4 a and b in the revised version). The statistics show that the report is unbiased and thus the dispersion or standard deviation can quantify the precision of the report. We followed the tradition by Bahrami et al. 2010 and defined the sensitivity as the inverse of the standard deviation of PDF in the revised version. Thus, a smaller standard deviation indicates a higher sensitivity, which further implies better performance.

How did the authors estimate sensitivity of a single participant? Using the data at the first round is not fair, as the sensitivity should be improved merely by the repetitive exposure to the same stimulus (i.e., even in individual perceptual decision-making, the sensitivity at the first round should be worse than that at the second and later rounds).

Reply: Thank you for the great question. The communication will affect the sensitivity from the second round on in the with-communication condition. Thus, we used the first-round reports as the approximations of the participants' individual estimations, which were not influenced by communication. Then, we calculated the standard deviation to estimate the sensitivity of a single participant.

In the without-communication condition, for trials with $\Delta = 0^\circ$ and no consensus formed, the performance (measured by the mean and standard error of means (s.e.m.) of

the report ($r-s$, report value minus the stimulus value) distribution) of a participant does not show consistent improvement as the task proceeds (Supplementary Fig. s2). Therefore, we think that it is acceptable to use the reports at the first round to approximate the single participant's estimates.

In the Bayesian integration models, how participants estimate their partner's confidence-level? Is it possible in this experimental paradigm (c.f., Bahrami et al., 2010)? The authors should provide the comprehensive descriptions about the models (not only for Bayesian models but also all of the other models).

Reply: We have provided detailed descriptions in the revised Supplementary Materials about all three models adopted in this study, including the virtual consensus by random sampling model (VCRS), the weighted confidence sharing model (WCS), and the direct signal sharing model (DSS).

In the WCS model proposed by Bahrami et al. 2010, the authors assumed that the participants communicated their confidence through free oral discussion. In that work, they applied a psychometric function $P(\Delta c) = H\left(\frac{\Delta c + b}{\sigma}\right)$ to measure performance, where Δc was the contrast difference, and $P(\Delta c)$ corresponded to the probability of saying that the second interval had the oddball with a contrast difference Δc , and $H(z) = \int_{-\infty}^z \frac{dt}{\sqrt{2\pi}} \exp[-t^2/2]$. They then assumed that confidence was represented by z-score $\Delta c/\sigma$, and the probability correct for the dyad was given by $P_{dyad}^{WCS}(\Delta c) =$

$\int_{\frac{x_1+x_2}{\sigma_1+\sigma_2} > 0} \frac{dx_1 dx_2}{2\pi\sigma_1\sigma_2} \exp\left[-\frac{(x_1-b_1-\Delta c)^2}{2\sigma_1^2} - \frac{(x_2-b_2-\Delta c)^2}{2\sigma_2^2}\right]$. After some calculation, they obtained

$b_{dyad}^{WCS} = \frac{s_1 b_1 + s_2 b_2}{s_1 + s_2}$ and $s_{dyad}^{WCS} = \frac{s_1 + s_2}{\sqrt{2}}$, where sensitivity $s = \frac{1}{\sqrt{2\pi}\sigma}$. For more details, please refer to the Supplementary Methods.

In our dyadic color estimation task, we did not measure the confidence of participants. The confidence may be indicated by how strongly a participant sticks to his own estimation. Here in this study, we just adopted the formulas from the Bahrami's research to make model predictions, which then could be compared to experimental results and predictions of other models, without adapting the derivation process.

The experimental task differs from other tasks used in the previous studies (e.g., Bahrami et al., 2010) in that pairs of the participants communicated with each other through observing their partner's previous choice. Does the difference in the experimental task accounts for the discrepancy in the findings between the present and the past studies?

Reply: Yes, we agree that the experimental design, especially the multi-round communication paradigm, leads to the discrepancy in the findings between the present and the previous studies. And we have provided more detailed explanations in section "Discussion" of the revised manuscript.

1) In the dyadic color estimation experiment, we varied the difference of stimulus and controlled the communicated information between participants, which helped us to understand the role of communication in making group consensus decision. In the previous studies, it is not easy to make sure how many rounds of communication happened and how much information has been communicated between participants [8-12]. However, in this study, we can easily compare the speed of the consensus formation due to the round-by-round fashion design. We showed that communication of estimates between participants speeds up the consensus formation, as evidenced by the observation that the probability of reaching a consensus within eight rounds of estimations with communication is significantly larger than that without communication (Fig. 2).

2) In our design, participants were requested to perceive and report the presented color. In the study by Bahrami et al. 2010, the participants were asked to perform a discrimination task. We had not manipulated the sensitivity of one participant in a dyad, but they manipulated individual sensitivity in a dyad by adding a substantial amount of

noise to their stimuli without having told the participants about this manipulation. Therefore, our data cannot discriminate the WCS model from the DSS model but their data can.

In our design, the participant would continue to report their estimates before the consensus formed within eight rounds of estimations, thus the consensus forced the participants simultaneously to make a good or bad choice, which improves the sensitivity of the dyad decision given normal participants. While in the study by Bahrami et al., the final collective decision was randomly chosen from one of the two members if the participants disagreed, which increases the randomness of the dyad choice and makes that dyad sensitivity cannot exceed that of the more sensitive participant.

3) To clarify the mechanism that consensus without communication outperforms that with communication, we can simply classify the estimates into good and bad ones. The good estimate is close to the stimulus and the bad estimate is far away from the stimulus. The probability of a bad consensus can be described as

$$P(b) = \frac{p(b_1, b_2)}{p(b_1, b_2) + p(g_1, g_2)}$$

where $p(g_1, g_2)$ is the probability that two participants simultaneously make good estimates, $p(b_1, b_2)$ is the probability that two participants simultaneously make bad estimates. For consensus with communication, estimates of one participant is dependent on the estimate of the other participant. We ignore the process of communication and simply apply the law of total probability, then we have:

$$P_w(b) = \frac{p(b_1)p(b_2|b_1)}{p(b_1)p(b_2|b_1) + p(g_1)p(g_2|g_1)} = \frac{p(b_1)}{p(b_1) + p(g_1) \frac{p(g_2|g_1)}{p(b_2|b_1)}}$$

where $p(g_2|g_1)$ is the conditional probability that one participant makes a good estimate given a good estimate by the other participant, $p(b_2|b_1)$ is the conditional probability that one participant makes a bad estimate given a bad estimate by the other participant. In our task, it is reasonable that the bad estimate far away from the stimulus has a stronger influence than the good estimate close to the stimulus to the estimate of the other participant, which implies that $\frac{p(g_2|g_1)}{p(g_2)} \leq \frac{p(b_2|b_1)}{p(b_2)}$. Thus, we can further obtain the result:

$$P_w(b) = \frac{p(b_1)}{p(b_1) + p(g_1) \frac{p(g_2|g_1)}{p(b_2|b_1)}} \geq \frac{p(b_1)}{p(b_1) + p(g_1) \frac{p(g_2)}{p(b_2)}} = \frac{p(b_1)p(b_2)}{p(b_1)p(b_2) + p(g_1)p(g_2)}$$

Because the participants make estimates independently in the without-communication condition, $\frac{p(b_1)p(b_2)}{p(b_1)p(b_2) + p(g_1)p(g_2)} = P_{w/o}(b)$ is the probability that a dyad reach a bad

consensus without communication. Therefore, the probability of bad consensus with communication is larger than that without communication, which indicates that the consensus without communication outperforms the consensus with communication.

Reviewer: 2

Comments to the Author(s)

Although, overall I find the paper interesting I have several concerns. Specifically, they are:

Introduction.

There are some important citations missing from the literature review. In particular, it would be nice to put the paper in the context of studies by Miller & Steyvers (2011); and Voinov, Sebanz, & Knoblich (2017) who worked in similar collective decisions in iterative interactions.

Reply: Thanks for your suggestions. We have enriched the whole manuscript, including section "Introduction". In the revised version, we introduced more background knowledge, including the two papers you mentioned, to better explain why we conducted this study (line 12-15).

Methods

Lines 35 - 38 are redundant and should be omitted.

Reply: This redundant paragraph has been deleted in the revised manuscript.

Line 57: the toolbox is titled "Psychtoolbox" and a proper reference to its website is missing.

Reply: Thanks for the suggestion. We have corrected it in the revised manuscript, and added a brief introduction to Psychtoolbox including its hyperlink (line 49-53).

Figure 1c caption: "Note that this trial-end feedback might be different for the two members due to the different stimuli they received". I didn't understand how two participants could receive different feedback. More detailed of the procedure in the text would help probably.

Reply: Sorry for the confusion. We have rephrased the section "Experimental procedure", and added more details to make it clear. The trial-end feedback includes three parts, the stimulus presented to him/her, his/her own report, and the report of the other participant in

a dyad. Because the presented stimulus may be different between participants, participants could receive different feedbacks. (Fig. 1c caption)

Line 21 - 22: This needs to follow immediately after the procedure of the "with-communication" condition to highlight the differences.

Reply: Thanks for the suggestion. We have rephrased the section "Experimental procedure", and added more details to make it clear (line 62-100).

In this study, we designed two main experimental conditions: with-communication condition and without-communication condition. In the first condition, the members in a dyad were allowed to communicate with each other through a black bar displayed on the color wheel in the report phase (Fig.1 d). In the second condition, the members in a dyad were not allowed to communicate and we did not show the estimate of other member in the report phase. Considering that different group members may get different information in the same decision-making scenarios in real life and that the difficulty to reach a consensus may depend on the difference between stimuli presented to participants, we further designed four conditions of stimulus difference wherein colors presented to two members in a dyad might be slightly different.In addition, ten randomly chosen dyads would receive the trial-end feedback information at the end of a trial in the with-communication condition (Fig. 1 c).

In the with-communication condition, a round started with presenting the round number and then a fixation marker on the screen for a brief period. Then a color disc (see the examples in Fig.1 a) was presented for one second and participants were requested to report their estimates by clicking on a color wheel (Figs. 1 d, e and f). Once the difference between the two participants' reports was smaller than a threshold (1° on the color wheel), the dyad reached a consensus and the trial ended. Otherwise, the trial moved to the next round. For each round in one trial, each participant was presented the same stimulus as in the first round. One trial would be terminated immediately after eight rounds of reports, regardless of whether a consensus has been reached or not. In the report phase of one round, each participant would see a black bar on the color wheel (Fig.1 b) indicating the estimate of his/her partner in the previous-round (Fig. 1 d), through which two members in a dyad communicated their estimates. That is, in the second and subsequent rounds in

one trial, each participant would see the same stimulus as in the first round and also be notified of his/her partner's previous-round estimate, indicating that one could adjust his/her estimate due to social influence in the with-communication condition.

In the without-communication condition, all the settings were the same as those in the with-communication condition with one exception. We did not show the previous estimate of one participant to another in the report phase. That is, participants would only see a color wheel with no black bar in the report phase, implying that participants could not communicate with each other.

Data Analysis

Line 58: Instead of multiple t-tests, the authors should run two-way ANOVA with Communication and Delta as factors. Also, t-tests in Supplementary Table 1 should be reported with p-values corrected for multiple comparisons.

Reply: Thanks for your helpful suggestion. We have conducted a two-factor mixed-design ANOVA based on the probability that a dyad reached a consensus within eight rounds of reports, to test the effects of communication and stimulus difference. The results revealed that communication between participants significantly speeds up consensus formation in the dyadic color estimation task, and the speed of consensus formation is not affected by stimulus difference in the with-communication condition. (line 144-160)

As for Supplementary Table 1 in the previous version (Supp. Tab. s2 in the new version), we have divided the comparisons into three subsets. We reported p-values corrected for multiple comparisons using Bonferroni correction in the new version. Please refer to the Supplementary Materials for more details.

Lines 9 - 18: My suggestion is to move the description of the model to the Supplementary Information, together with specifications of the WCS and DSS models (which are currently only mentioned briefly). Regarding the VCRS model, if I understood it correctly, it sampled from discrete distributions built from individual independent responses of the the dyad members. Wouldn't a proper model of the process sample from continuous distributions (viz., Gaussian), estimated from individual response data? After all, participants' responses are made on a continuous scale, which

makes a consensus by chance so difficult to emerge. Is this the reason why the VCRS model underestimates the rate of trials which never reached a consensus in the without-communication condition?

Reply: Thanks for the suggestions. We moved the details of the WCS, DSS, and VCRS model to the Supplementary Materials. The VCRS model mimics the consensus formation process in the without-communication condition by randomly sampling two reports from two participants' first-round report collections.

In the previous version of manuscript, the probability that the sampled reports would reach a consensus in the n -th step was described as $p(n) = p_1(1 - p_1)^{n-1}$, where p_1 was approximated as the proportion of trials reaching a consensus in the first round of reports. This formula overestimated the probability of consensus from Round 2 to Round 7, thus underestimated the rate of trials which did not reach a consensus in the without-communication condition.

In the revised version of manuscript, we directly sampled from the collection of participant's reports to form the consensus, instead of approximating that by the above formula. The new results better explain the experimental data in the without-communication condition, especially when $\Delta = 0^\circ$ (see Fig. 2c). However, the new version underestimates the probability of reaching a consensus within 8 rounds, thus overestimates the ratio of trials that did not reach a consensus within 8 rounds of estimations. A possible reason may be that two individuals' reports with respect to the same stimulus are not completely random, but with some correlation. (The fourth row in Supp. Fig. 3 gives a clue. However, this proposal certainly needs more investigations in future.)

We also sampled from the continuous Gaussian distribution estimated from individual report data, and found that the rate of trials reaching a consensus within 8 rounds would be further underestimated, as shown in the following figure. We think the reason may be that, the reports were not truly continuous due to the color wheel was composed of 360 colors and the reports were discrete.

Figures 4a and 4b "Dyad collection" are missing P1 curves.

Reply: Actually, P1 curve and P2 curve were almost overlapped in the right panels of Fig. 4a and 4b in the previous manuscript. "Dyad collection" was the collection of all dyads, thus the statistical properties were almost the same for two virtual collected participants. Therefore, P1 curve and P2 curve (Gaussian PDF) are still very close in Fig. 4a and 4b in the new manuscript.

Figure 4d. I am not convinced that the inverse U function is the best fit to the data, and its interpretations are based on a rather weak evidence. If the authors want to make a claim that on the relationship between RoI and Improvement Ratio is not just linear, but that a quadratic component of the model explains more variation in the data, they need to make a proper model comparison of the log-likelihood values.

Reply: Thanks for your helpful suggestion. We applied the maximum likelihood estimation to find the parameters of the fitting function. We also provided the log likelihood value for the linear fitting function, $L = 23.577$ and the quadratic fitting function, $L = 28.672$. A further likelihood ratio test using the *lratiotest* function in Matlab showed that a quadratic function

better interpreted the data than a linear function ($\chi^2(1) = 10.189, p = 0.001$). This result suggests that dyads with a moderate RoI improved the consensus more than dyads with an RoI that was either too small or too large, implying that dyads in which members are willing to listen to others but not with complete obedience are apt to reach a better consensus. Certainly, more evidence is needed to support this implication. (line 209-214)

Lines 57 - 62. Participants' performance in the no-communication condition, as well as predictions from the VCRS model, was more accurate than the optimal Bayesian model. How can this be? My understanding is that the DSS and WCS models take as an input only one observation (that is, from the first round of a trial), while the VCRS model makes use of multiple observations through consecutive rounds. In other words, the algorithm collects multiple samples which results in a more reliable integration. All this needs to be spelled out in the text.

Reply: Yes, participants' performance in the without-communication condition was more accurate than the optimal Bayesian model, which is surprising to us at first. We agree with that multiple rounds of viewing stimulus and estimations play key roles here. The consensus through the consecutive reports itself can improve the performance by decreasing the probability of making mistakes. To detail this, we can simply classify the estimates into good and bad ones. The good estimate is close to the stimulus and the bad estimate is far away from the stimulus. The probability of a bad consensus can be described as

$$P(b) = \frac{p(b_1, b_2)}{p(b_1, b_2) + p(g_1, g_2)}$$

where $p(g_1, g_2)$ is the probability that two participants simultaneously make good estimates, while $p(b_1, b_2)$ is the probability that two participants simultaneously make bad estimates. For consensus with communication, estimates of one participant is dependent on the estimate of the other participant. Using the law of total probability, we can obtain:

$$P_w(b) = \frac{p(b_1)}{p(b_1) + p(g_1) \frac{p(g_2|g_1)}{p(b_2|b_1)}}$$

Where $p(g_2|g_1)$ is the conditional probability that one participant makes a good estimate given that the other participant makes a good estimate. $p(b_2|b_1)$ is the conditional probability that one participant makes a bad estimate given that the other participant makes

a bad estimate. In our task, it is reasonable that the bad estimate far away from the stimulus has a stronger influence than the good estimate close to the stimulus to the estimate of the other participant, which implies that $\frac{p(g_2|g_1)}{p(g_2)} \leq \frac{p(b_2|b_1)}{p(b_2)}$. Thus, we further obtain the result:

$$P_w(b) = \frac{p(b_1)}{p(b_1) + p(g_1) \frac{p(g_2|g_1)}{p(b_2|b_1)}} \geq \frac{p(b_1)}{p(b_1) + p(g_1) \frac{p(g_2)}{p(b_2)}} = \frac{p(b_1)p(b_2)}{p(b_1)p(b_2) + p(g_1)p(g_2)}$$

Because the participants make estimates independently in the without-communication condition, $\frac{p(b_1)p(b_2)}{p(b_1)p(b_2) + p(g_1)p(g_2)} = P_{w/o}(b)$ is the probability that a dyad reach a bad consensus without communication. Therefore, the probability of bad consensus with communication is larger than that without communication, which indicates that the consensus without communication outperforms the consensus with communication. Moreover, the consensus without communication also outperforms the predictions of the WCS model and the DSS model, which are not significantly different from the consensus with communication. A possible reason may be that the latter two models just integrate one round of information.

A general question on the "estimations quality" data. How was data aggregated when consensus was not reached? Were two judgments from the last round averaged or was an average accuracy of the two judgments taken? This needs to be specified explicitly.

Reply: We specified the information of these questions in the revised manuscript in section "Data analyses" (line 117-122).

1) When we analyzed the effects of communication on performance quality, we only took trials with consensus formed and $\Delta = 0^\circ$ into consideration.

2) A consensus was calculated as the mean of the two members' last-round reports, between which the difference was smaller than 1° . For trials with $\Delta \neq 0^\circ$, one cannot measure their qualities, because no correct answer exists.

3) Trials with no consensus formed were also excluded from the performance quality analyses.

Discussion

The most important aspect not touched in the discussion, is why dyads performed more accurate in the no - communication condition, and why they outperformed a Bayesian optimal algorithm. The authors provide no psychological interpretation for this interesting finding, and the reader is left to wonder why one mode of interaction turns out to be more effective than another. The exact mechanism underlying the effect needs to be spelled out here.

Reply: Thanks for the helpful comments. We have rewritten the related paragraph in section “Discussion” and explained why the consensus without communication outperforms that with communication and predictions of the WCS and DSS model (line 257-273).

The paper would benefit from discussing some broader implications of the findings, including non-perceptual decision-making problems. In which situations collective estimations benefit from communication and when is communication detrimental? The conclusions made, e.g. "However, it might be beneficial that group members share information about the problem before making their own decisions to gather as much information as possible" are not related to the results.

Reply: Thanks for the helpful suggestions. We have rewritten and extended the discussion.

1) We have discussed why consensus without communication outperforms that with communication (line 257-273).

2) We discussed the mechanism underlying the speed of consensus formation (line 232-247).

3) We also discussed more implications of our findings on non-perceptual decision-making problems as below:

Although these results were obtained through a perceptual dyadic decision-making task where all information of the stimuli had been provided to each participant, these results could provide insights into non-perceptual group decision-making scenarios. First, speed-accuracy tradeoff has been often observed in group or individual decision-making and the possible mechanisms have been proposed [32–34]. In this study, we also observed a similar phenomenon, namely, the consensus without communication takes longer time but

has a higher precision, while the consensus with communication takes shorter time but has a lower precision. Second, for a non-perceptual group decision-making scenario with all information to the members of the group, which is similar to our experiment, a better group consensus can be reached after multiple rounds of independent decision of each member. Third, in some non-perceptual group decision-making scenarios, information is incomplete and communication can provide additional information. The members possess more information or more resources usually have more influence than others when making group decisions. The problem is how to effectively allocate a proper weight to each member in this type of group decision-making. Obviously, this is beyond our experiment, but our study may provide some insights into this problem. For example, we split the decision into two stages. In the first stage, members anonymously communicate with each other to dig out as much information as possible, which can weaken the influence of one member on other members during communication. In the second stage, multiple rounds of independent decisions can form a better consensus as in our experiment. Of course, this problem perhaps deserves more detailed investigations in future.

The biggest problem with the manuscript, thought, is its language and multiple grammatical and stylistic errors throughout the paper. The authors are advised to have their manuscript proof-read, if possible, by a native English speaker, before they submit the final version.

Reply: Thank you for your advice. We have carefully checked the grammar in the revised manuscript. One of our friends who is a senior lecturer in Ulster University had proofread our manuscript and polished the language.

Appendix C

Dear Editors and Reviewers,

Thank you for the helpful comments and suggestions. We have revised our manuscript according to reviewers' comments. The details are as follows.

- (1) In the "*Discussion*" section, we have moved the technical details to the *Supplementary Information*, and have carefully checked the grammar and syntax of this section, to make it concise and clear.
- (2) Other minor revisions are shown below.

Thank you again for the helpful comments and suggestions. Please find below the point-by-point reply.

Best wishes,

Yours sincerely,

DaHui Wang

Reviewer comments to Author:

Reviewer: 1

Comments to the Author(s)

I acknowledge the authors' efforts to revise the manuscript. I appreciate it. However, I still have a couple of concerns.

[Major concerns]

My critical concern is about the way of the data analysis. As far as I understand, what the authors actually did is (1) to get the original data; (2) to get the additional data; and (3) to analyze the original and the additional data together. The sample size (the number of participants in the original + that in the additional data) was determined by a power analysis based on the effect size estimated from the original data. This procedure was not consistent with the standard way of hypothesis testing. In other words, pilot data for a power analysis to determine sample size should be independent from the main data for testing the hypothesis. I believe what the authors should have done is (1) to get the original data; (2) to conduct a power analysis based on the original data in order to determine sample size of additional data; (3) to get the additional data; (3) to analyze the additional data separated from the original data.

Reply: Thank you for your question. The sample size was determined by a power analysis based on the effect size estimated from a previously published study about collective perceptual decision-making (Bahrami et al., 2010). (Line 44-47) Both the original and additional data were independent from the power analysis. Therefore, it is reasonable to combine the original data and additional data for analyses.

If I understand correctly, the present study suggests that people are more likely to reach a consensus in the presence of communication, compared with the case in the absence of communication. On the other hand, if we focus ONLY on the cases in which the pair of participants reached a consensus, the dyadic consensus estimates in the absence of communication outperformed those in the presence of communication. Given that, I believe the claim that consensus estimation without communication generally outperforms that with communication is an overstatement.

Reply: Thanks for the comment. Firstly, our study suggests that in this specific dyadic color estimation experiment, communication helps two participants in a dyad to reach a consensus more easily and quickly. Under the constrain of eight rounds, the mean number (across dyads) of trials with consensus formed is 65.3 (or 49.4) in the with-communication (or without-communication) condition when $\Delta = 0^\circ$ (see the following figure), providing enough samples for a statistical test. Thus, for the trials with consensus reached when $\Delta = 0^\circ$, consensus reports in the without-communication condition are better than that in the with-communication condition in a statistical sense. Therefore, we argued that communication speeds up but impairs the consensus decision in such a dyadic color estimation task, which is a low-level perceptual decision-making task. There is no doubt that this result cannot be directly applied to other scenarios, as previous studies have already shown that whether communication is beneficial or detrimental actually depends on the context, which implies that more studies are needed to further explore the effects of communication on group decision-making. The highlight of the current study is the experimental design used to explore the role of communication in the dynamic process of consensus formation and its quality, where task-relevant information each member possesses and the round-by-round communicated information are controllable. Due to its simplicity and controllability, this paradigm can possibly be extended to explore the cognitive mechanisms and neural substrates underlying dyadic decision-making or even large-scale group decision-making.

[Minor concern]

In Figure 4c, it would be interesting to show the data about the "bias" not only the "sensitivity". While the authors mentioned that the bias was not significantly different from zero, there might be differences between the conditions.

Reply: Thanks for the comment. We have reported results about the “bias” in the *Supplementary Information* (see Supp. Fig. s4 and Supp. Tab. s1). The results show that μ is not significantly different from zero for the distribution of the single participant’s reports or that of dyadic consensus reports with or without communication, indicating that the single participant’s estimates and the dyadic consensus estimates are unbiased (Line 183-186). Only the result of the VCRS model is slightly larger than 0° , but smaller than 1° , i.e. the threshold of consensus formation.

	P1	P2	With comm.	W/o comm.	VCRS	WCS	DSS
h	0	0	0	0	1	0	0
t_{stat}	-1.286	-0.722	-2.206	0.404	3.064*	-0.909	-0.704
p	1	1	0.263	1	0.025	1	1

Reviewer: 2

Comments to the Author(s)

Most of the issues I raised have been properly addressed in the revised version. Some minor issues would still need to be addressed though (see below):

Lines 50 - 53 reads as if it was an ad, it would be better to remove it.

Reply: Thanks for the suggestion. We have removed this sentence in the revised manuscript.

Line 74 - 75" In each condition of with-communication and without-communication, participants accomplished about 200 trials in total"

I didn't understand why the authors didn't provide the exact number of trials. Was it different for different dyads? If so, why was there variable amount of trials in different conditions for different dyads?

Reply: Thanks for the question. Through some preliminary tests, we have observed that one dyad needs to spend about 10 minutes to finish a block, which contains 10 trials. Therefore, we offered each dyad four hours in total (including between-block rest) and expected them to accomplish 200 trials. However, the amount of trials accomplished by different dyads are different because they execute the task with different reaction time. The amount of trials each dyad accomplished are in the range of 160~240, shown in the following figure. Moreover, for each dyad, the ratio between four stimulus differences (0° , 3° , 6° , 9°) is almost unaffected (0.4: 0.25: 0.25: 0.1).

The term "Dyad Collection" is an improper one. Probably, the authors meant "Aggregated across Dyads" or simply "Averaged".

Reply: Thanks for the suggestion. We have replaced “*Dyad collection*” with “*Aggregation across dyads*” in the revised manuscript and relevant figures.

Unnumbered Lines between Line 257 and 258. In general, I find it not a so good idea to fill the Discussion section with technical details unless they are essential to the final conclusions (which is not the case here). The manuscript would benefit if the authors move the formulae from this section either to the Supplementary Information or to the Appendix and rephrase their explanation in a less technical, compressed, way.

Reply: Thanks for the suggestion. We have moved the technical details to the *Supplementary Information*, and tried to interpret the mechanism in a more compressed way in the *Discussion* section.

I also found that, although the rest of the manuscript has been proof-read and so now it reads nice and neat, the new parts of the Discussion section are still replete with syntax errors. It is very desirable to have another round of proof-reading for this specific part.

Reply: Thanks for the advice. We have carefully checked and revised the grammar and syntax in the *Discussion* section, and proofread the whole manuscript again.